# Adaptively Exploiting $d$-Separators with Causal Bandits

**Blair Bilodeau**[*]
University of Toronto

**Linbo Wang**
University of Toronto

**Daniel M. Roy**
University of Toronto

## Abstract

Multi-armed bandit problems provide a framework to identify the optimal intervention over a sequence of repeated experiments. Without additional assumptions, minimax optimal performance (measured by cumulative regret) is well-understood. With access to additional observed variables that $d$-separate the intervention from the outcome (i.e., they are a *d-separator*), recent "causal bandit" algorithms provably incur less regret. However, in practice it is desirable to be agnostic to whether observed variables are a $d$-separator. Ideally, an algorithm should be *adaptive*; that is, perform nearly as well as an algorithm with oracle knowledge of the presence or absence of a $d$-separator. In this work, we formalize and study this notion of adaptivity, and provide a novel algorithm that *simultaneously* achieves (a) optimal regret when a $d$-separator is observed, improving on classical minimax algorithms, and (b) significantly smaller regret than recent causal bandit algorithms when the observed variables are not a $d$-separator. Crucially, our algorithm does not require any oracle knowledge of whether a $d$-separator is observed. We also generalize this adaptivity to other conditions, such as the front-door criterion.

## 1 Introduction

Given a set of interventions (actions) for a specific experiment, we are interested in learning the best one with respect to some outcome of interest. Without knowledge of specific causal structure relating the observed variables, this task is impossible from solely observational data (c.f. Theorem 4.3.2 of [29]). Instead, we seek the most efficient way to sequentially choose interventions for i.i.d. repetitions of the experiment, where the main challenge is that we cannot observe the counterfactual effect of interventions we did not choose. Without any structural assumptions beyond i.i.d., one can always learn the best intervention with high confidence by performing each intervention a sufficient number of times [10]. In the presence of additional structure—such as a causal graph on observables—this strategy may result in performing suboptimal interventions unnecessarily often. However, the presence of such structure is often unverifiable, and incorrectly supposing that it exists may catastrophically mislead the experimenter. Thus, a fundamental question arises: Can we avoid strong, unverifiable assumptions while simultaneously performing fewer harmful interventions when advantageous structure exists?

A natural framework in which to study this question is that of (multi-armed) bandit problems: Over a sequence of interactions with the environment, the experimenter chooses an action using their experience of the previous interactions, and then observes the *reward* of the chosen action. The goal is to achieve comparable performance with what would have been achieved if the experimenter had chosen the (unknown) optimal action in each interaction. Formally, performance is measured by *regret*, which is the difference of the *cumulative reward* incurred by the experimenter compared to the optimal action. In this partial-information setting, regret induces the classical trade-off between *exploration* (choosing potentially suboptimal actions to learn if they're optimal) and *exploitation*

---

[*]Correspondence to blair.bilodeau@mail.utoronto.ca

36th Conference on Neural Information Processing Systems (NeurIPS 2022).

(choosing the action that empirically appears the best). In contrast, other measures of performance (e.g., only identifying the average treatment effect or the best action at the end of all interactions) do not penalize the experimenter for performing suboptimal actions during exploration, and consequently are insufficient to study our question of interest.

For an *action set* $\mathcal{A}$ and a *time horizon* $T$, the minimax optimal regret for bandits without any assumptions on the data (worst-case) is $\tilde{\mathcal{O}}(\sqrt{|\mathcal{A}|\,T})$ [6], and is achieved by many algorithms [19]. Recently, Lu et al. [24] showed that, under additional causal structure, a new algorithm (C-UCB) can achieve improved regret. In particular, if the experimenter has access to a variable $Z$—taking values in a finite set $\mathcal{Z}$—that *d-separates* [29] the intervention and the reward, as well as the interventional distribution of $Z$ for each $a \in \mathcal{A}$, C-UCB achieves $\tilde{\mathcal{O}}(\sqrt{|\mathcal{Z}|\,T})$ regret. However, as we show, the performance of C-UCB when the $d$-separation assumption fails is orders of magnitude worse than that of UCB. Is *strict adaptation* possible? That is, is there an algorithm that recovers the guarantee of C-UCB when $Z$ is a $d$-separator and the guarantee of UCB in all other environments, without advance knowledge of whether $Z$ is a $d$-separator?

As of yet, there is no general theory of adaptivity in the bandit setting. The closest we have to a general method is the Corral algorithm and its offspring [2, 4]. Corral uses online mirror descent to combine "base" bandit algorithms, but requires that each of the base algorithms is "stable" when operating on importance-weighted observations. Unfortunately, while UCB is stable, simulations reveal this not to be the case for C-UCB. This presents a barrier to adapting to causal structure via Corral-like techniques, and raises the question of whether there is a new way to achieve adaptivity.

**Contributions.** We introduce the *conditionally benign* property for bandit environments: informally, there exists a random variable $Z$ such that the conditional distribution of the reward given $Z$ is the same for each action $a \in \mathcal{A}$. We show that the conditionally benign property is (a) strictly weaker than the assumption of Lu et al. [24] (in their proofs, they actually assume that all causal parents of the reward are observed); (b) equivalent to $Z$ being a $d$-separator when $\mathcal{A}$ is all interventions; and (c) implied by the front-door criterion [29] when $\mathcal{A}$ is all interventions except the *null intervention* (i.e., a pure observation). We then prove that any algorithm that achieves optimal worst-case regret must incur suboptimal regret for some conditionally benign environment, and hence strict adaptation to the conditionally benign property is impossible. Despite this, we introduce the hypothesis-tested adaptive causal upper confidence bound algorithm (HAC-UCB), which provably (a) achieves non-vacuous (sublinear in $T$) regret at all times without any assumptions, (b) recovers the improved performance of C-UCB for conditionally benign environments, and (c) performs as well as UCB in certain environments where C-UCB and related algorithms (such as those studied in [25, 27]) incur linear regret. Empirically, we observe these performance improvements on simulated data.

**Impact.** Recently, multiple works have developed causal bandit algorithms that achieve improved performance in the presence of advantageous causal relationships (initiated by Bareinboim et al. [7] and Lattimore et al. [17]; see Section 7 for more literature). Further, the last decade has seen a flurry of work in bandits on designing algorithms that recover worst-case regret bounds while simultaneously performing significantly better in advantageous settings, without requiring advance knowledge of which case holds [e.g., 9, 31, 34, 26, 1]. However, to the best of our knowledge, no existing work studies algorithms that achieve *adaptive regret guarantees with respect to causal structure*. The present work provides a framework that expands the study of adaptive decision making to the rich domain of causal inference.

## 2 Preliminaries

### 2.1 Problem Setting

We consider a general extension of the usual bandit setting where, in addition to a reward corresponding to the action played, the experimenter observes some additional variables *after* choosing their action; we call this the *post-action context*. This is distinct from the contextual bandit problem, where the experimenter has access to side-information *before* choosing their action.

Let $\mathcal{Y} = [0, 1]$ be the *reward space*[2], $\mathcal{Z}$ be a finite set of values for the post-action context to take, and $\mathscr{P}(\mathcal{Z} \times \mathcal{Y})$ denote the set of joint probability distributions. For any $p \in \mathscr{P}(\mathcal{Z} \times \mathcal{Y})$ and

---

[2]Our results hold for $\mathcal{Y} = \mathbb{R}$ using sub-Gaussian rewards with bounded mean at the expense of constants.

$(\mathcal{Z}, \mathcal{Y})$-valued random variable $(Z, Y)$, let $p(Z)$ and $p(Y \mid Z)$ denote the marginal and conditional distributions respectively. Let $\mathbb{E}_p$ and $\mathbb{P}_p$ denote expectation and probability operators under $p$.

The *stochastic bandit problem with post-action contexts* proceeds as follows: For each round $t \in [T]$, the experimenter selects $A_t \in \mathcal{A}$ while simultaneously $\{(Z_t(a), Y_t(a)) : a \in \mathcal{A}\}$ are independently sampled from the *environment*, which is any family of distributions $\nu = \{\nu_a : a \in \mathcal{A}\} \in \mathscr{P}(\mathcal{Z} \times \mathcal{Y})^{\mathcal{A}}$ indexed by the action set. The experimenter only observes $(Z_t(A_t), Y_t(A_t))$ and receives reward $Y_t(A_t)$. From a causal perspective, $(Z_t(a), Y_t(a))_{a \in \mathcal{A}}$ corresponds to the potential outcome vector, and under causal consistency, $(Z_t(A_t), Y_t(A_t))$ corresponds to the observed data $(Z_t, Y_t)$ under the chosen intervention $A_t$.

The *observed history up to round* $t$ is the random variable $H_t = (A_s, Z_s(A_s), Y_s(A_s))_{s \in [t]}$. A *policy* is a sequence of measurable maps from the observed history to the action set, denoted

$$\pi = (\pi_t)_{t \in [T]} \in \Pi(\mathcal{A}, \mathcal{Z}, T) := \prod_{t=1}^{T} \left\{ (\mathcal{A} \times \mathcal{Z} \times \mathcal{Y})^{t-1} \to \mathcal{A} \right\}.$$

The experimenter chooses a policy in advance to select their action on each round according to $A_t = \pi_t(H_{t-1})$. Clearly, an environment $\nu$ and a policy $\pi$ together define a joint probability distribution on $(A_t, Z_t(a), Y_t(a))_{t \in [T], a \in \mathcal{A}}$ (which includes the "counterfactuals" not seen by the player). Let $\mathbb{E}_{\nu,\pi}$ denote expectation under this joint distribution. The performance of a policy under an environment is quantified by the *regret*

$$R_{\nu,\pi}(T) = T \cdot \max_{a \in \mathcal{A}} \mathbb{E}_{\nu_a}[Y] - \mathbb{E}_{\nu,\pi} \sum_{t=1}^{T} \mathbb{E}_{\nu_{A_t}}[Y].$$

## 2.2 Specific Algorithms

Classical bandit algorithms take $|\mathcal{A}|$ and $T$ as inputs. The dependence on $T$ can often be dropped using the doubling trick or more sophisticated techniques such as decreasing learning rates, but we do not focus on these refinements in this work, instead allowing $T$ as an input. In order to account for the additional information provided by the post-action context, we also consider algorithms that take $|\mathcal{Z}|$ as an input. By restricting dependence to only the cardinality of $\mathcal{A}$ and $\mathcal{Z}$, we explicitly suppose that there is no additional structure to exploit on these spaces; much work in the bandit literature has focused on such structure through linear or Lipschitz rewards, but we defer these extensions to future work in favour of focusing on adaptivity. For notational simplicity, we denote algorithmic dependence on $\mathcal{A}$ or $\mathcal{Z}$ even though the dependence is actually through their cardinality (i.e., the labellings of items in the sets are arbitrary).

In the causal bandit literature [17, 24, 27], it is common to suppose that the algorithm also receives distributional information relating actions to intermediate variables. In particular, if the (unknown) environment is $\nu$, prior work supposes that the algorithm has access to the (interventional) marginal distributions $\nu(Z) = \{\nu_a(Z) : a \in \mathcal{A}\}$. In this work, we suppose instead that the algorithm has access to a collection of *approximate* marginal distributions $\tilde{\nu}(Z) = \{\tilde{\nu}_a(Z) : a \in \mathcal{A}\}$; for example, these could be an estimate of $\nu(Z)$ that was learned offline. Ideally, $\tilde{\nu}(Z)$ will be close to $\nu(Z)$, but our novel method is entirely adaptive to this assumption: regardless of how well $\tilde{\nu}(Z)$ approximates $\nu(Z)$, HAC-UCB incurs sublinear regret.

We now introduce additional notation to define the algorithms of interest in this work. Suppose that $\mathcal{A}$, $\mathcal{Z}$, $\tilde{\nu}(Z)$, and $T$ are all fixed in advance, as well as a confidence parameter $\delta = \delta_T \in (0, 1)$. For each $t \in [T]$, $z \in \mathcal{Z}$, and $a \in \mathcal{A}$, define the number of the first $t$ rounds on which $z$ was observed by $\mathbb{T}_t^{\mathcal{Z}}(z) = 1 \vee \sum_{s=1}^{t} \mathbb{I}\{Z_s(A_s) = z\}$ (where $a \vee b = \max\{a, b\}$), and similarly the number of rounds on which $a$ was chosen by $\mathbb{T}_t^{\mathcal{A}}(a) = 1 \vee \sum_{s=1}^{t} \mathbb{I}\{A_s = a\}$. Further, define the empirical mean estimate for the reward under the distribution induced by choosing the action $a$ as $\hat{\mu}_t^{\mathcal{A}}(a) = [\mathbb{T}_t^{\mathcal{A}}(a)]^{-1} \sum_{s=1}^{t} Y_s(A_s) \mathbb{I}\{A_s = a\}$ and the empirical conditional mean estimate for the reward given that $z$ was observed as $\hat{\mu}_t^{\mathcal{Z}}(z) = [\mathbb{T}_t^{\mathcal{Z}}(z)]^{-1} \sum_{s=1}^{t} Y_s(A_s) \mathbb{I}\{Z_s(A_s) = z\}$. Define $\mathrm{UCB}_t^{\mathcal{A}}(a) = \hat{\mu}_t^{\mathcal{A}}(a) + \sqrt{\log(2/\delta)/(2\mathbb{T}_t^{\mathcal{A}}(a))}$, $\mathrm{UCB}_t^{\mathcal{Z}}(z) = \hat{\mu}_t^{\mathcal{Z}}(z) + \sqrt{\log(2/\delta)/(2\mathbb{T}_t^{\mathcal{Z}}(z))}$, and $\widehat{\mathrm{UCB}}_t(a) = \sum_{z \in \mathcal{Z}} \mathrm{UCB}_t^{\mathcal{Z}}(z) \mathbb{P}_{\tilde{\nu}_a}[Z = z]$.

Using these objects, we define three algorithms, each of which produces actions that are $H_t$-measurable. The upper confidence bound algorithm (UCB, [6]) is defined by $A_{t+1}^{\mathrm{UCB}} =$

$\arg\max_{a\in\mathcal{A}}\mathrm{UCB}_t^{\mathcal{A}}(a)$, and the causal upper confidence bound algorithm (C-UCB, [24]) is defined by $A_{t+1}^{\mathrm{C}}=\arg\max_{a\in\mathcal{A}}\widetilde{\mathrm{UCB}}_t(a)$, where ties are broken by using some predetermined ordering on $\mathcal{A}$. Finally, we define a new combination of these two methods, which we call the hypothesis-tested adaptive causal upper confidence bound algorithm (HAC-UCB) and describe precisely in Algorithm 1; we denote its actions by $A_{t+1}^{\mathrm{HAC}}$.

---

**Algorithm 1:** HAC-UCB($\mathcal{A}$, $\mathcal{Z}$, $T$, $\tilde{\nu}(Z)$)

---

**do** Play each $a\in\mathcal{A}$ for $\lceil 4\sqrt{T}/|\mathcal{A}|\,\rceil$ rounds, and let $\hat{\nu}_a(Z)$ be the MLE of $\nu_a(Z)$

**if** $\sup_{a\in\mathcal{A}}\sum_{z\in\mathcal{Z}}\left|\mathbb{P}_{\tilde{\nu}_a}[Z=z]-\mathbb{P}_{\hat{\nu}_a}[Z=z]\right|>2T^{-1/4}\sqrt{|\mathcal{A}|\,|\mathcal{Z}|\log T}$

$\quad\mid\quad$ **replace** $\tilde{\nu}(Z)\longleftarrow\hat{\nu}(Z)$

**do** Play each $a\in\mathcal{A}$ for $\lceil\sqrt{T}/|\mathcal{A}|\,\rceil$ rounds

**set** flag = True

**while** $t\le T$

$\quad\mid\quad$ **if** flag

$\qquad\mid\quad$ /* Check if either of the two conditions fail $\qquad\qquad\qquad\qquad$ */

$\qquad\mid\quad$ **set** $\mathrm{D}_{t-1}^{\mathcal{A}}(a)=\mathrm{UCB}_{t-1}^{\mathcal{A}}(a)-\widetilde{\mathrm{UCB}}_{t-1}(a)+\frac{\sqrt{|\mathcal{A}||\mathcal{Z}|\log T}}{T^{1/4}}$

$\qquad\mid\quad$ **for** $a\in\mathcal{A}$ **do**

$\qquad\qquad\mid\quad$ **if not** $-2\sum_{z\in\mathcal{Z}}\sqrt{\frac{\log T}{\mathbb{T}_{t-1}^{\mathcal{Z}}(z)}}\mathbb{P}_{\tilde{\nu}_a}[Z=z]\le\mathrm{D}_{t-1}^{\mathcal{A}}(a)\le 2\sqrt{\frac{\log T}{\mathbb{T}_{t-1}^{\mathcal{A}}(a)}}+2\frac{\sqrt{|\mathcal{A}||\mathcal{Z}|\log T}}{T^{1/4}}$

$\qquad\qquad\mid\quad\mid\quad$ **set** flag = False; **break**

$\qquad\mid\quad$ /* If conditions pass, play C-UCB $\qquad\qquad\qquad\qquad\qquad\qquad\qquad$ */

$\qquad\mid\quad$ **if** flag

$\qquad\qquad\mid\quad$ **set** $A_t^{\mathrm{HAC}}=A_t^{\mathrm{C}}$;

$\qquad\mid\quad$ **else**

$\qquad\qquad\mid\quad$ **set** $A_t^{\mathrm{HAC}}=A_t^{\mathrm{UCB}}$;

$\quad\mid\quad$ **else**

$\qquad\mid\quad$ /* If conditions ever fail, play UCB forever $\qquad\qquad\qquad\qquad$ */

$\qquad\mid\quad$ **set** $A_t^{\mathrm{HAC}}=A_t^{\mathrm{UCB}}$;

---

Heuristically, HAC-UCB has an initial exploration period to ensure that $\tilde{\nu}(Z)$ is sufficiently accurate—if not, it is replaced with the maximum likelihood estimate (MLE) of the marginals—and then optimistically plays C-UCB until there is sufficient evidence that the environment is not conditionally benign. The switch from C-UCB to UCB is decided by a hypothesis test performed on each round, which uses the confidence intervals that will hold if the environment is conditionally benign. When the arm mean estimates of UCB and C-UCB disagree, this provides evidence that the environment is not conditionally benign, and the evidence is considered sufficient to switch when the size of the disagreement is large compared to the size of the confidence intervals themselves. As we illustrate in the proof of the regret bounds, with high probability this test will not induce a switch for a conditionally benign environment, and will sufficiently limit the regret incurred by C-UCB if the environment is not conditionally benign.

## 3 Conditionally Benign Property

We now formalize the main property that HAC-UCB will adaptively exploit.

**Definition 3.1.** *An environment* $\nu\in\mathscr{P}(\mathcal{Z}\times\mathcal{Y})^{\mathcal{A}}$ *is conditionally benign if and only if there exists* $p\in\mathscr{P}(\mathcal{Z}\times\mathcal{Y})$ *such that for each* $a\in\mathcal{A}$, $\nu_a(Z)\ll p(Z)$ *and* $\nu_a(Y\mid Z)=p(Y\mid Z)$ *p-a.s.*

This definition does not require any causal terminology to define or use for regret bounds, but we now instantiate it for the causal setting. For a collection of finite random variables $\mathbf{V}$ and a (potentially) continuous random variable $Y$, let $\mathscr{P}_{\mathbf{V}}$ be the set of all joint probability distributions with strictly positive marginal probabilities on $\mathbf{V}$. Fix a DAG $\mathcal{G}$ on $(\mathbf{V},Y)$ such that $Y$ is a leaf and two disjoint sets $\mathbf{Z}\subseteq\mathbf{V}$ and $\mathbf{A}\subseteq\mathbf{V}$ such that $\mathrm{Pa}_{\mathbf{A}}^{\mathcal{G}}\subseteq\mathbf{V}\setminus\mathbf{Z}$. Let $\mathcal{A}$ be the set of all possible $\mathrm{do}$ interventions on $\mathbf{A}$, and for each $a\in\mathcal{A}$ let $p_a$ denote the interventional distribution (Definition A.5). This structure suggests a graphical analogue of the conditionally benign property.

**Definition 3.2.** *For any DAG $\mathcal{G}$ and $\mathcal{A}' \subseteq \mathcal{A}$, $(\mathcal{G}, \mathcal{A}')$ is conditionally benign if and only if for all $p \in \mathscr{P}_{\mathbf{V}}$ that are Markov relative to $\mathcal{G}$, $\{p_a(\mathbf{Z}, Y) : a \in \mathcal{A}'\}$ is conditionally benign.*

We now connect the conditionally benign property to $d$-separation (Definition A.6) and the front-door criterion (Definition A.8). All proofs are deferred to Appendix A, along with standard notation and definitions from the causal literature.

**Theorem 3.3.** $\mathbf{Z}$ *$d$-separates $Y$ from $\mathbf{A}$ on $\mathcal{G}$ if and only if $(\mathcal{G}, \mathcal{A})$ is conditionally benign.*

This equivalence is a strict specialization of the conditionally benign property to the causal setting. In particular, to define conditionally benign, we need not require all possible interventions be allowed. Define $\mathcal{A}_0$ to be $\mathcal{A}$ with the null (observational) intervention removed, and let $\mathcal{G}_{\bar{\mathbf{A}}}$ denote $\mathcal{G}$ with all edges directed into $\mathbf{A}$ removed.

**Theorem 3.4.** $\mathbf{Z}$ *$d$-separates $Y$ from $\mathbf{A}$ on $\mathcal{G}_{\bar{\mathbf{A}}}$ if and only if $(\mathcal{G}, \mathcal{A}_0)$ is conditionally benign.*

The benefits of discarding the null intervention are demonstrated by the following fact.

**Lemma 3.5.** $\mathbf{Z}$ *$d$-separates $Y$ from $\mathbf{A}$ on $\mathcal{G}_{\bar{\mathbf{A}}}$ if $\mathbf{Z}$ satisfies the front-door criterion for $(\mathbf{A}, Y)$ on $\mathcal{G}$.*

We visualize the preceding results in Figure 1. In graph (a), $Z$ $d$-separates the intervention from the reward, and hence any Markov relative distribution (Definition A.3) will induce a conditionally benign environment. Graph (b) corresponds to a setting where one cannot hope to always improve performance due to the direct effect of the intervention on the reward, and consequently the environment need not be conditionally benign. In graph (c), the presence of the unobserved confounder $U$ means that $Z$ does not $d$-separate the intervention from the reward. However, if the null intervention is not considered, the arrow from $U$ to $A$ is never applicable, and hence any Markov relative distribution on the modified DAG will induce a conditionally benign environment. Specifically, graph (c) satisfies the front-door criterion, revealing that the conditionally benign property captures that this setting is still benign for decision-making, even though the conditions assumed by Lu et al. [24] do not hold. Finally, in graph (d), the unobserved confounder $U$ once again violates $d$-separation, but also the front-door criterion is not satisfied because of the back-door path from $Z$ to $Y$. Hence, even discarding the null intervention does not guarantee that the environment will be conditionally benign.

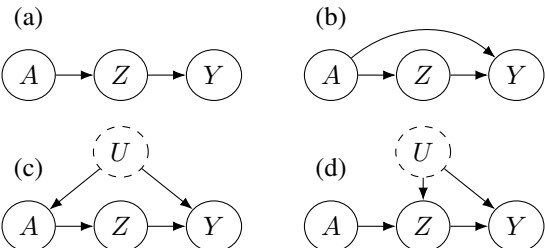

Figure 1: DAGs to illustrate the conditionally benign property. $A$ is the intervention, $Z$ is the post-action context, $Y$ is the reward, and $U$ is an unobserved variable. $(\mathcal{G}, \mathcal{A})$ is conditionally benign for (a) but only $(\mathcal{G}, \mathcal{A}_0)$ is conditionally benign for (c). For (b) and (d) the environment need not be conditionally benign.

## 4 Analysis of Bandit Algorithms

We now study the impact of the conditionally benign property on regret. All proofs are deferred to Appendix B. First, recall the standard regret bound for UCB, with constants tuned to rewards in $[0, 1]$.

**Theorem 4.1** (Theorem 7.2 of [19])**.** *For all $\mathcal{A}$, $\mathcal{Z}$, $T$, and $\nu \in \mathscr{P}(\mathcal{Z} \times \mathcal{Y})^{\mathcal{A}}$, if $\delta = 2/T^2$*

$$R_{\nu,\text{UCB}}(T) \le 2\,|\mathcal{A}| + 4\sqrt{2\,|\mathcal{A}|\,T \log T}.$$

Second, we generalize the main result of Lu et al. [24] by relaxing two assumptions: we only require that the environment is conditionally benign, and we allow for approximate marginal distributions using the following definition. Later, this will enable us to trade-off approximation error of $\tilde{\nu}(Z)$ with online estimation of $\nu(Z)$ in order to ensure that HAC-UCB *always* incurs sublinear regret.

**Definition 4.2.** *For any $\varepsilon \geq 0$, $\tilde{\nu}(Z)$ and $\nu(Z)$ are $\varepsilon$-close if*

$$\sup_{a \in \mathcal{A}} \sum_{z \in \mathcal{Z}} \left| \mathbb{P}_{\tilde{\nu}_a}[Z = z] - \mathbb{P}_{\nu_a}[Z = z] \right| \leq \varepsilon.$$

**Theorem 4.3** (Refined Theorem 1 of [24]). *For all $\varepsilon > 0$, $\mathcal{A}$, $\mathcal{Z}$, $T$, and conditionally benign $\nu \in \mathscr{P}(\mathcal{Z} \times \mathcal{Y})^{\mathcal{A}}$, if $\tilde{\nu}(Z)$ and $\nu(Z)$ are $\varepsilon$-close and $\delta = 2/T^2$ then*

$$R_{\nu,\mathrm{c}}(T) \leq 2\,|\mathcal{Z}| + 6\sqrt{|\mathcal{Z}|\,T\log T} + (\log T)\sqrt{2T} + 2\varepsilon(1 + \sqrt{\log T})T.$$

**Remark 4.4.** *Lu et al. [24] assume that $\varepsilon = 0$. Our result implies that an approximation error of $\varepsilon = \sqrt{|\mathcal{Z}|/T}$ is sufficient to achieve the optimal rate.* ◁

Next, we motivate our introduction of a new algorithm by showing that C-UCB can catastrophically fail when the environment is *not* conditionally benign, incurring regret that is linear in $T$ (which is as bad as possible for bounded rewards).

**Theorem 4.5.** *For every $\mathcal{A}$ and $\mathcal{Z}$ with $|\mathcal{A}| \geq 2$, there exists $\nu \in \mathscr{P}(\mathcal{Z} \times \mathcal{Y})^{\mathcal{A}}$ such that even if $\tilde{\nu}(Z) = \nu(Z)$, for all possible settings of the confidence parameters, $\delta_T$,*

$$\lim_{T \to \infty} \frac{R_{\nu,\mathrm{c}}(T)}{T} \geq 1/120.$$

**Remark 4.6.** *This lower bound is specifically for C-UCB. However, any algorithm that relies on eliminating actions from consideration via assumption rather than data is susceptible to such an issue. In particular, this result is easily modified to apply for C-TS [24] and C-UCB-2 [27]. Other causal algorithms (e.g., Parallel Bandit [17]) also intuitively suffer from the issue that our construction exploits, although a different argument must be made since these rely on different causal structure.* ◁

We now state our regret upper bound for our new algorithm, HAC-UCB. In Theorem 6.2, we will show it is *impossible* to always achieve the optimal regret without knowledge of whether a $d$-separator is observed, but the following theorem shows *some* adaptivity is always possible. Crucially, HAC-UCB achieves sublinear regret without any assumptions on $\nu$ or $\tilde{\nu}(Z)$. For a more detailed breakdown of the constants, see Eq. (B.9).

**Theorem 4.7** (Main Result). *For all $\mathcal{A}$, $\mathcal{Z}$, $T \geq 25\,|\mathcal{A}|^2$, $\nu \in \mathscr{P}(\mathcal{Z} \times \mathcal{Y})^{\mathcal{A}}$, and $\tilde{\nu}(Z) \in \mathscr{P}(\mathcal{Z})^{\mathcal{A}}$,*

$$R_{\nu,\mathrm{HAC}}(T) \leq 4\,|\mathcal{A}| + 11\,T^{3/4}(\log T)\sqrt{|\mathcal{A}|\,|\mathcal{Z}|} + 15\sqrt{(|\mathcal{A}| + |\mathcal{Z}|)T\log T} + 5(\log T)\sqrt{T}.$$

*For all $\varepsilon \leq T^{-1/4}\sqrt{|\mathcal{A}|\,|\mathcal{Z}|\log T}$, if $\nu$ is conditionally benign and $\tilde{\nu}(Z)$ and $\nu(Z)$ are $\varepsilon$-close then*

$$R_{\nu,\mathrm{HAC}}(T) \leq 4\,|\mathcal{A}| + 2\,|\mathcal{Z}| + 6\sqrt{|\mathcal{Z}|\,T\log T} + 4(\log T)\sqrt{T} + 2\varepsilon(1 + \sqrt{\log T})T.$$

It is an open problem whether the dependence on $T^{3/4}$ is tight. In Theorem 6.2, we will show that it is impossible to obtain worst-case regret of size $\sqrt{|\mathcal{A}|\,T}$ while still achieving improved regret on conditionally benign environments. However, it may be possible to improve the dependence on $T$, and the role of logarithmic factors in how much improvement is possible remains to be understood. Towards improving this result, we now show that there exists an environment that forces C-UCB to incur linear regret yet HAC-UCB will switch to following UCB (and hence incurs $\sqrt{|\mathcal{A}|\,T\log T}$ regret at worst). That is, HAC-UCB recovers the improved performance of C-UCB when the conditionally benign property holds, is never worse than C-UCB, and optimally outperforms C-UCB in some settings.

**Theorem 4.8.** *There exists a constant $C$ such that for any $\mathcal{A}$ and $\mathcal{Z}$ with $|\mathcal{A}| \geq 2$, there exists $\nu \in \mathscr{P}(\mathcal{Z} \times \mathcal{Y})^{\mathcal{A}}$ so that for any $\delta_T$ used for C-UCB with $\tilde{\nu}(Z) = \nu(Z)$,*

$$\lim_{T \to \infty} \frac{R_{\nu,\mathrm{c}}(T)}{T} \geq 1/C,$$

*and if $\tilde{\nu}(Z) = \nu(Z)$ is used for HAC-UCB then*

$$\lim_{T \to \infty} \frac{R_{\nu,\mathrm{HAC}}(T)}{|\mathcal{A}| + |\mathcal{Z}| + \sqrt{|\mathcal{A}|\,T\log T} + (\log T)\sqrt{T}} \leq C.$$

**Remark 4.9.** *Theorem 4.8 could be stated with $\tilde{\nu}(Z)$ only $\varepsilon$-close to $\nu(Z)$, but for simplicity we have supposed $\varepsilon = 0$ to highlight the role of the conditionally benign property.* ◁

# 5 Simulations

We now study the empirical performance of these algorithms in two key settings, corresponding to a conditionally benign environment and the lower bound environment from Theorem 4.5. We compare our algorithm HAC-UCB with UCB [5], C-UCB [24], C-UCB-2 [27], and Corral (for which we use the learning rate and algorithm prescribed by [2] with the epoch-based, scale-sensitive UCB prescribed by [4]); for all algorithms, we use the parameters that are optimal as prescribed by existing theory. To focus solely on the impact of the conditionally benign property, we set $\tilde{\nu}(Z) = \nu(Z)$. The results of this section are a representative simulation demonstrating empirically that (a) for worst-case environments, both C-UCB and C-UCB-2 incur linear regret, while HAC-UCB successfully switches to incur sublinear regret to compete with Corral and UCB, and (b) for conditionally benign environments, HAC-UCB and C-UCB enjoy improved performance compared to UCB, Corral, and C-UCB-2, all three of which have regret growing like $\sqrt{|\mathcal{A}| T}$. Implementation details are available in Appendix D and code can be found at https://github.com/blairbilodeau/adaptive-causal-bandits.

## 5.1 Conditionally Benign Environment

First, we consider a conditionally benign environment. Taking the gap $\Delta = \sqrt{|\mathcal{A}| (\log T)/T}$, the fixed conditional distribution for $\mathcal{Z} = \{0, 1\}$ is $Y \mid Z \sim \text{Ber}(1/2 + (1 - Z)\Delta)$. Then, for a small $\varepsilon$ (we take $\varepsilon = 0.0005$), we set $\mathbb{P}_{\nu_1}[Z = 0] = 1 - \varepsilon$ and $\mathbb{P}_{\nu_a}[Z = 0] = \varepsilon$ for all other $a \in \mathcal{A} \setminus \{1\}$. Thus, $a_\nu^* = 1$, and the actions are separated by $\Delta$. In summary, each $z \in \mathcal{Z}$ has positive probability of being observed, yet each action nearly deterministically fixes $Z$.

In Figure 2 (left panel) we observe three main effects: (a) C-UCB and HAC-UCB perform similarly (their regret curves overlap), both achieving much smaller regret that remains unchanged by increasing $|\mathcal{A}|$, (b) UCB grows at the worst-case rate of roughly $\sqrt{|\mathcal{A}| T \log T}$, not taking advantage of the conditionally benign property, and (c) neither Corral nor C-UCB-2 realize the benefits of the conditionally benign property, since the regret increases with $|\mathcal{A}|$ and empirically they perform worse than UCB. We note that the x-axis starts at $T = 500$ to satisfy the minor requirement of $T > |\mathcal{A}|^2$.

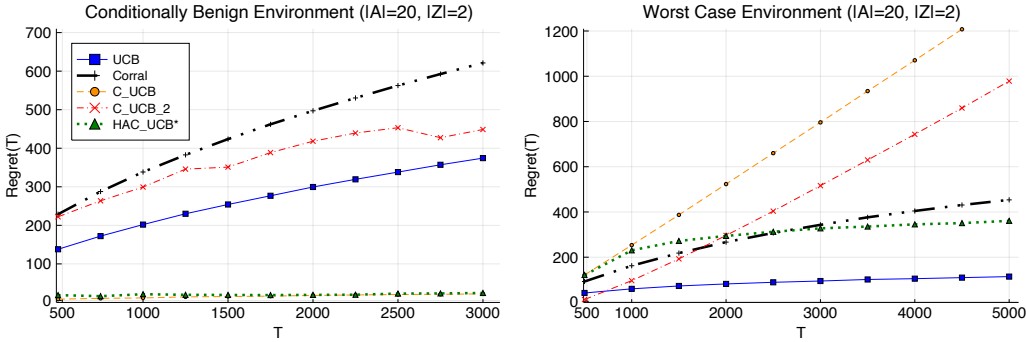

Figure 2: Regret when the conditionally benign property holds (left) and when it fails (right).

## 5.2 Worst Case Environment

Second, we consider an environment that is *not* conditionally benign. We use the same general environment from our lower bound in Theorem 4.5, where the causal algorithms learn a biased estimate of which $z \in \{0, 1\}$ has a higher conditional mean since $Z$ is not a $d$-separator, and consequently they concentrate onto a bad action.

In Figure 2 (right panel), we again observe three main effects: (a) both C-UCB and C-UCB-2 incur linear regret, as prescribed by Theorem 4.5, (b) HAC-UCB achieves sublinear regret, although it is worse than UCB (we show in Theorem 6.2 that optimal adaptivity is impossible), and (c) Corral does not suffer linear regret, but appears to still do worse than HAC-UCB for large $T$.

# 6 Adaptivity for Causal Structures

Thus far we have analyzed the regret of various algorithms in two cases: environments that either are or are not conditionally benign. Minimax algorithms (UCB) fail to achieve smaller regret for conditionally benign environments, while causal algorithms (C-UCB) fail catastrophically in environments that are not conditionally benign. In this section, we formalize the notion of strict adaptivity (*adaptive minimax optimality*) with respect to the conditionally benign property, show that it is *impossible* to be adaptively minimax optimal with respect to the conditionally benign property, and discuss a relaxed notion of optimal adaptivity based on Pareto optimality.

## 6.1 Generic Algorithms

In order to describe adaptivity (and its impossibility) in the stochastic bandit problem with post-action contexts, we require a higher level of abstraction than a policy, which we achieve with *algorithms*. It is possible to define algorithms and the corresponding notion of adaptivity in abstract generality. However, we take the same perspective as Bilodeau et al. [8] (who define adaptive minimax optimality with respect to relaxations of the i.i.d. assumption), and sacrifice some generality by defining algorithms using the specific objects we study in this work. Formally, an algorithm is any map from the problem-specific inputs to the space of compatible policies, denoted by

$$\mathfrak{a} : (\mathcal{A}, \mathcal{Z}, T, \tilde{\nu}(Z)) \mapsto \mathfrak{a}(\mathcal{A}, \mathcal{Z}, T, \tilde{\nu}(Z)) \in \Pi(\mathcal{A}, \mathcal{Z}, T).$$

We denote the set of all algorithms by $\mathbb{A}_{\text{C-MAB}}$, and the subset of algorithms that are constant in $(\mathcal{Z}, \tilde{\nu}(Z))$ by $\mathbb{A}_{\text{MAB}}$; this subset contains the classical bandit algorithms that are agnostic to knowledge of post-action contexts, or more specifically, do not exploit causal structure.

## 6.2 Adaptive Minimax Optimality

Let $\mathfrak{p} : \mathscr{P}(\mathcal{Z} \times \mathcal{Y})^{\mathcal{A}} \mapsto \{0, 1\}$ encode whether a given environment satisfies a certain property of interest; in this work, it is always an indicator for whether $\nu$ is conditionally benign. Further, for every $q \in \mathscr{P}(\mathcal{Z})$, denote the set of all environments with marginal $q$ by $\Pi_{\mathcal{A}, \mathcal{Z}}(q) = \{\nu \in \mathscr{P}(\mathcal{Z} \times \mathcal{Y})^{\mathcal{A}} : \nu(Z) = q\}$. There are multiple ways one could define optimal adaptivity: we propose the following notion of strict adaptivity, which requires that the experimenter to do as well as they possibly could have if they had access to $\mathfrak{p}(\nu)$ in advance, but without this knowledge.

**Definition 6.1.** *An algorithm* $\mathfrak{a} \in \mathbb{A}_{\text{C-MAB}}$ *is* adaptively minimax optimal *with respect to* $\mathfrak{p}$ *if and only if there exists $C > 0$ such that for all $\mathcal{A}$, $\mathcal{Z}$, $q \in \mathscr{P}(\mathcal{Z})^{\mathcal{A}}$, and $T$,*

$$\sup_{\nu \in \Pi_{\mathcal{A}, \mathcal{Z}}(q)} R_{\nu, \mathfrak{a}(\mathcal{A}, \mathcal{Z}, q, T)}(T) \leq C \inf_{\pi \in \Pi(\mathcal{A}, \mathcal{Z}, T)} \sup_{\nu \in \Pi_{\mathcal{A}, \mathcal{Z}}(q)} R_{\nu, \pi}(T) \tag{6.1}$$

*and*

$$\sup_{\nu \in \Pi_{\mathcal{A}, \mathcal{Z}}(q), \, \mathfrak{p}(\nu)=1} R_{\nu, \mathfrak{a}(\mathcal{A}, \mathcal{Z}, q, T)}(T) \leq C \inf_{\pi \in \Pi(\mathcal{A}, \mathcal{Z}, T)} \sup_{\nu \in \Pi_{\mathcal{A}, \mathcal{Z}}(q), \, \mathfrak{p}(\nu)=1} R_{\nu, \pi}(T). \tag{6.2}$$

## 6.3 Impossibility of Strict Adaptivity

We now show it is *impossible* for any algorithm to always realize the benefits of the conditionally benign property while also recovering the worst-case rate of $\sqrt{|\mathcal{A}| \, T}$ (e.g., Theorems 9.1 and 15.2 of [19]), even when the algorithm has access to the true marginals. Our proof strategy is a modification of the finite-time lower bounds from Section 16.2 of Lattimore and Szepesvári [19]. Notably, the lower bounds of Lu et al. [24] already imply that any algorithm that does not take advantage of causal structure cannot be adaptively minimax optimal. We prove a significantly stronger result: even algorithms that use $\mathcal{Z}$ and $\tilde{\nu}(Z) = \nu(Z)$ cannot be adaptively minimax optimal!

**Theorem 6.2.** *Let $\mathfrak{a} \in \mathbb{A}_{\text{C-MAB}}$ be such that there exists $C > 0$ such that for all $\mathcal{A}$, $\mathcal{Z}$, and $T$,*

$$\sup_{\nu \in \mathscr{P}(\mathcal{Z} \times \mathcal{Y})^{\mathcal{A}}} R_{\nu, \mathfrak{a}(\mathcal{A}, \mathcal{Z}, \nu(Z), T)}(T) \leq C \sqrt{|\mathcal{A}| \, T}.$$

*There exists a constant $C'$ such that for all $\mathcal{A}$, $\mathcal{Z}$, and $T \geq |\mathcal{A}|$, there exists conditionally benign $\nu \in \mathscr{P}(\mathcal{Z} \times \mathcal{Y})^{\mathcal{A}}$ with*

$$R_{\nu, \mathfrak{a}(\mathcal{A}, \mathcal{Z}, \nu(Z), T)}(T) \geq C' \sqrt{|\mathcal{A}| \, T}.$$

### 6.4 Pareto-Adaptive Minimax Optimality

In light of this impossibility result, it is of interest to characterize relaxations of strict adaptivity that are achievable. Koolen [15] and Lattimore [18] introduce the *Pareto frontier* of regret, which when applied to the conditionally benign property, is all tuples of regret guarantees such that improving the regret in conditionally benign environments would necessarily force the worst-case regret to increase. We propose that it is desirable for an algorithm to do as well as possible in the worst-case, subject to always realizing smaller regret on conditionally benign environments. Formally, let $\mathbb{A}^\star_{\text{C-MAB}}$ be the subset of $\mathfrak{a} \in \mathbb{A}_{\text{C-MAB}}$ that satisfy Eq. (6.2) for some constant $C$.

**Definition 6.3.** *An algorithm* $\mathfrak{a}^\star \in \mathbb{A}^\star_{\text{C-MAB}}$ *is* Pareto-adaptively minimax optimal *with respect to* $\mathfrak{p}$ *if and only if there exists* $C > 0$ *such that for all* $\mathfrak{a} \in \mathbb{A}^\star_{\text{C-MAB}}$, $\mathcal{A}$, $\mathcal{Z}$, $q \in \mathscr{P}(\mathcal{Z})^{\mathcal{A}}$, *and* $T$,

$$\sup_{\nu \in \Pi_{\mathcal{A},\mathcal{Z}}(q)} R_{\nu,\mathfrak{a}^\star(\mathcal{A},\mathcal{Z},q,T)}(T) \leq C \sup_{\nu \in \Pi_{\mathcal{A},\mathcal{Z}}(q)} R_{\nu,\mathfrak{a}(\mathcal{A},\mathcal{Z},q,T)}(T).$$

It remains an open problem to prove whether HAC-UCB is Pareto-adaptively minimax optimal, and more generally to identify the Pareto frontier for the causal bandit problem.

## 7 Related Work

Kocaoglu et al. [14] and Lindgren et al. [23] efficiently used interventions to learn a causal graph under standard causal assumptions [29]. Hyttinen et al. [13] identified the minimal number of experiments needed to learn underlying causal structure of variables with only a linear structural assumption. Lee and Bareinboim [20, 22, 21] identified the minimal set of interventions that permit causal effect identification in the presence of known causal structure, while Kumor et al. [16] studied analogues of the *back-door condition* for identifying causal effects with unobserved confounders.

Bareinboim et al. [7] introduced causal bandits with a binary motivating example to demonstrate that empirically better performance can be achieved by exploiting a specific, known causal structure. Lattimore et al. [17] and Yabe et al. [35] studied best-arm identification, where the experimenter does not incur any penalty for exploration rounds. Given knowledge of the causal graph informing the interventions and response, they separately proved that exponential improvements in the dependence of the regret on the action set are possible provided the underlying distribution on the causal graph is sufficiently "balanced".

Sen et al. [32] obtained an instance-dependent regret bound under causal assumptions, but obtained the wrong dependence on the arm gap ($\Delta^{-2}$ rather than $\Delta^{-1}$), and consequently in the worst-case the dependence on $T$ may still dwarf the structural benefits. Sen et al. [33] studied an alternative type of intervention, where rather than fixing a node only the conditional probabilities are changed. This notion is easily stated in our notation, since we allow for abstract families of distributions (indexed by abstract "interventions") to define a environment. However, they focused on distribution-dependent guarantees under stronger causal assumptions, and hence our results are not directly comparable.

All of the above regret bounds heavily require assumptions about the causal graph, and without such assumptions the presumed information learned from non-intervening rounds can catastrophically mislead the experimenter in exactly the same way that C-UCB suffers in our Theorem 4.5. Hence, it remains an interesting open problem to study adaptivity in each of these variations of the causal bandit setting, and our work provides a stepping stone to do so.

Prior to the present work, Lu et al. [24] has already been extended in multiple directions. de Kroon et al. [12] observed that C-UCB can be reduced to requiring only a separating set, but only prove the regret is no worse than that of UCB if a separating set is observed. The authors remark that a causal discovery algorithm could in principle be used to learn the separating set online, but observed in their experiments that they obtain biased estimates and hence there are no convergence guarantees. Lu et al. [25] replaced knowledge of the causal graph with the assumption that the causal graph has a tree structure, and incorporated the cost of learning the tree into the full regret bound. Nair et al. [27] provided an instance-dependent regret bound for an alternative algorithm to C-UCB, which they call C-UCB-2, in the presence of the full causal graph. While they demonstrated empirically that C-UCB-2 outperforms C-UCB for certain instances, we find that C-UCB-2 performs much worse when a $d$-separator is observed, and the provable linear lower bound (Theorem 4.5) also applies to C-UCB-2 when there are no observed $d$-separators.

# 8 Discussion

We have demonstrated that the improved regret possible when a $d$-separator is observed can also be realized in the multi-armed bandit problem by requiring only certain conditional independencies, which we have formalized using the conditionally benign property. We proved that it is impossible to optimally adapt to this property, but provided a new algorithm (HAC-UCB) that simultaneously recovers the improved regret for conditionally benign environments and significantly improves on prior work when the conditionally benign property does not hold. Crucially, our algorithm requires no more assumptions about the world than vanilla bandit algorithms. We expect our results to spur future work on (a) improved adaptation to the conditionally benign property, (b) relaxations of the conditionally benign property for which optimal adaptation is possible, and (c) adaptation in more general partial feedback settings.

In practice, HAC-UCB will be most useful in settings with a large action space and intermediate variables that may plausibly satisfy the conditionally benign property. In passing, we mention that one such example is learning the causal effect of genome edits (interventions) on disease phenotypes. Here, the post-action context could be gene expressions that are sometimes assumed to be a $d$-separator (e.g., [3]). We leave the implementation of our algorithm in clinical settings and collaboration with practitioners for future work.

## Acknowledgements

BB is supported by an NSERC Canada Graduate Scholarship and the Vector Institute. DMR is supported in part by an NSERC Discovery Grant, Canada CIFAR AI Chair funding through the Vector Institute, and an Ontario Early Researcher Award. This material is based also upon work supported by the United States Air Force under Contract No. FA850-19-C-0511. Any opinions, findings and conclusions or recommendations expressed in this material are those of the author(s) and do not necessarily reflect the views of the United States Air Force. Part of this work was done while BB and DMR were visiting the Simons Institute for the Theory of Computing. We thank Zachary Lipton for suggesting to consider simultaneously adapting to estimation of the marginal distributions, and we thank Chandler Squires, Vasilis Syrgkanis, and Angela Zhou for helpful discussions.

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
