# A Proofs for Causal Equivalences

We begin with a restating of standard definitions for completeness.

## A.1 Standard Results from Causal Literature

To be more explicit about the role of $Y$ in the causal setting (namely, it may be continuous), we introduce some more notation. Let $\mathbf{V} = (V_1, \ldots, V_M, Y)$ denote random variables each taking values in $\mathcal{V}_i = \{v_i^1, \ldots, v_i^{k_i}\}$ and $[0, 1]$ respectively. A *causal bandit graph* is any directed acyclic graph (DAG) $\mathcal{G}$ defined on the nodes $(V_1, \ldots, V_M, Y)$ such that (a) $Y$ is a leaf node and (b) if there is a directed arrow from $V_i$ to $V_{i'}$, then $i < i'$. Let $\mathscr{P}_{\mathbf{V}}$ denote the set of all probability distributions $p$ on $(V_1, \ldots, V_M, Y)$ such that the marginal probabilities over $(V_1, \ldots, V_M)$ are all strictly positive.

**Definition A.1** (Markovian parents, Definition 1.2.1 of [29]). *For any $p \in \mathscr{P}_{\mathbf{V}}$ and $i \in [M]$, the Markovian parents of $V_i$ under $p$ is the minimum-cardinality subset $\mathbf{V}' \subseteq (V_1, \ldots, V_{i-1})$ such that $V_i \perp (V_1, \ldots, V_{i-1}) \setminus \mathbf{V}' \mid \mathbf{V}'$ under $p$. We denote this by $\mathrm{Pa}_i^p$. Similarly, $\mathrm{Pa}_Y^p$ is the minimum-cardinality subset $\mathbf{V}' \subseteq \mathbf{V}$ such that $Y \perp \mathbf{V} \setminus \mathbf{V}' \mid \mathbf{V}'$ under $p$.*

**Definition A.2** (Graphical parents). *For any causal bandit graph $\mathcal{G}$, the graphical parents of $V_i$ under $\mathcal{G}$ is the unique subset $\mathbf{V}' \subseteq \mathbf{V}$ of variables that have a directed arrow into $V_i$. We denote this by $\mathrm{Pa}_i^{\mathcal{G}}$. Similarly, $\mathrm{Pa}_Y^{\mathcal{G}}$ is the unique subset $\mathbf{V}' \subseteq \mathbf{V}$ of variables that have a directed arrow into $Y$.*

**Definition A.3** (Markov relative, Theorem 1.2.7 of [29]). *A distribution $p \in \mathscr{P}_{\mathbf{V}}$ is Markov relative to a causal bandit graph $\mathcal{G}$ if any only if for all $V \in \mathbf{V} \cup \{Y\}$, $\mathrm{Pa}_V^p \subseteq \mathrm{Pa}_V^{\mathcal{G}}$.*

**Remark A.4** (Equation 1.33 of [29]). *If $p \in \mathscr{P}_{\mathbf{V}}$ is Markov relative to $\mathcal{G}$, then for all measurable $\mathcal{B} \subseteq \mathcal{Y}$ and $(j_1, \ldots, j_M) \in \prod_{i=1}^M [k_i]$,*

$$p(Y \in \mathcal{B}, V_1 = v_1^{j_1}, \ldots, V_M = v_M^{j_M}) = p(Y \in \mathcal{B} \mid \mathrm{Pa}_Y^{\mathcal{G}} = \mathbf{u}_Y) \prod_{i=1}^M p(V_i = v_i^{j_i} \mid \mathrm{Pa}_i^{\mathcal{G}} = \mathbf{u}_i),$$

*where the conditioning is understood to be on the event where the parents take the specific values defined by $\mathbf{u} = (v_1^{j_1}, \ldots, v_M^{j_M})$.* ◁

**Definition A.5** (Causal intervention, Definition 1.3.1 of [29]). *Let $p \in \mathscr{P}_{\mathbf{V}}$ be Markov relative to $\mathcal{G}$. The interventional distribution induced on $\mathbf{V}$ by the intervention $a = \mathrm{do}(V_{i_1} = v_{i_1}^{j_{i_1}}, \ldots, V_{i_\ell} = v_{i_\ell}^{j_{i_\ell}})$ is*

$$p_a(Y \in \mathcal{B}, V_1 = v_1^{j_1}, \ldots, V_M = v_M^{j_M})$$
$$= \mathbb{I}\{V_{i_1} = v_{i_1}^{j_{i_1}}, \ldots, V_{i_\ell} = v_{i_\ell}^{j_{i_\ell}}\} p(Y \in \mathcal{B} \mid \mathrm{Pa}_Y^{\mathcal{G}} = \mathbf{u}_Y) \prod_{i \notin \{i_1, \ldots, i_\ell\}} p(V_i = v_i^{j_i} \mid \mathrm{Pa}_i^{\mathcal{G}} = \mathbf{u}_i).$$

**Definition A.6** (*d*-Separated, Definition 2.4.1 of [30]). *$\mathbf{Z}$ $d$-separates $Y$ from $\mathbf{A}$ (on $\mathcal{G}$) if and only if every path between $\mathbf{A}$ and $Y$ is blocked; that is, every path contains either (a) $\bigcirc \to B \to \bigcirc$ or $\bigcirc \leftarrow B \to \bigcirc$ such that $B \in \mathbf{Z}$, or (b) $\bigcirc \to B \leftarrow \bigcirc$ with no descendents of $B$ (including itself) in $\mathbf{Z}$.*

**Definition A.7** (Back-Door Path, Section 3.3.1 of [29]). *A path from $\mathbf{Z}$ to $\mathbf{Z}'$ is a back-door path if it begins with an arrow directed into $\mathbf{Z}$.*

**Definition A.8** (Front-Door Criterion, Definition 3.3.3 of [29]). *$\mathbf{Z}$ satisfies the front-door criterion relative to $(\mathbf{A}, Y)$ on $\mathcal{G}$ if and only if (a) all directed paths from $\mathbf{A}$ to $Y$ pass through $\mathbf{Z}$, (b) there is no unblocked back-door path from $\mathbf{A}$ to $\mathbf{Z}$, and (c) all back-door paths from $\mathbf{Z}$ to $Y$ are blocked by $\mathbf{A}$.*

## A.2 Proof of Theorem 3.3

We first state an intuitive result about $d$-separation that is used often in the causal literature, but we could not find stated or proved precisely as follows.

**Lemma A.9.** *If $\mathbf{Z}$ $d$-separates $Y$ from $\mathbf{A}$ and $\mathrm{Pa}_{\mathbf{A}}^{\mathcal{G}} \subseteq \mathbf{V} \setminus \mathbf{Z}$, then $\mathbf{Z}$ $d$-separates $Y$ from $(\mathbf{A}, \mathrm{Pa}_{\mathbf{A}}^{\mathcal{G}})$.*

*Proof of Lemma A.9.* First, every path from $\mathrm{Pa}_{\mathbf{A}}^{\mathcal{G}}$ to $Y$ that passes through $\mathbf{A}$ must satisfy one of (a) or (b) since a subpath does. Further, every path from $\mathrm{Pa}_{\mathbf{A}}^{\mathcal{G}}$ to $Y$ that doesn't pass through $\mathbf{A}$ can be

extended to a (back-door) path from $\mathbf{A}$ to $Y$ using the edge $\mathrm{Pa}_{\mathbf{A}}^{\mathcal{G}} \to \mathbf{A}$, but $\mathrm{Pa}_{\mathbf{A}}^{\mathcal{G}} \subseteq \mathbf{V} \setminus \mathbf{Z}$ (and hence this part of the path cannot satisfy either property), so the original path must satisfy either (a) or (b). $\qquad \square$

Next, we recall the probabilistic equivalence of $d$-separation.

**Theorem A.10** (Theorem 1.2.4 of [29]). *Fix a causal bandit graph $\mathcal{G}$ and two disjoint sets $\mathbf{Z} \subseteq \mathbf{V}$ and $\mathbf{A} \subseteq \mathbf{V}$ such that $\mathrm{Pa}_{\mathbf{A}}^{\mathcal{G}} \subseteq \mathbf{V} \setminus \mathbf{Z}$. Then $\mathbf{Z}$ $d$-separates $Y$ from $\mathbf{A}$ if and only if $Y \perp \mathbf{A} \mid \mathbf{Z}$ under every distribution $p \in \mathscr{P}_{\mathbf{V}}$ that is Markov relative to $\mathcal{G}$.*

We now turn to the main argument to prove Theorem 3.3. Let $p \in \mathscr{P}_{\mathbf{V}}$ and $a = \mathrm{do}(\mathbf{A} = \mathbf{a})$ be arbitrary. First, we prove the "Causal Effect Rule" [30]. For any $\mathbf{V}' \subseteq \mathbf{V}$,

$$p_a(\mathbf{V}' = \mathbf{v}') = \sum_{\mathbf{u}} p_a(\mathbf{V}' = \mathbf{v}' \mid \mathrm{Pa}_{\mathbf{A}}^{\mathcal{G}} = \mathbf{u}) p_a(\mathrm{Pa}_{\mathbf{A}}^{\mathcal{G}} = \mathbf{u})$$

$$= \sum_{\mathbf{u}} p(\mathbf{V}' = \mathbf{v}' \mid \mathbf{A} = \mathbf{a}, \mathrm{Pa}_{\mathbf{A}}^{\mathcal{G}} = \mathbf{u}) p(\mathrm{Pa}_{\mathbf{A}}^{\mathcal{G}} = \mathbf{u}),$$

where the sum is over all possible values that $\mathrm{Pa}_{\mathbf{A}}^{\mathcal{G}}$ can take and we have used that (a) conditional on $\mathrm{Pa}_{\mathbf{A}}^{\mathcal{G}}$, the interventional and conditional distributions given $\mathbf{A} = \mathbf{a}$ are equivalent, and (b) the marginal distribution of $\mathrm{Pa}_{\mathbf{A}}^{\mathcal{G}}$ is unchanged by intervening on $\mathbf{A}$.

Now, suppose $\mathbf{Z}$ $d$-separates $Y$ from $\mathbf{A}$. Then, it follows that

$$
\begin{aligned}
p_a(Y \in \mathcal{B} \mid \mathbf{Z} = \mathbf{z}) &= \frac{p_a(Y \in \mathcal{B}, \mathbf{Z} = \mathbf{z})}{p_a(\mathbf{Z} = \mathbf{z})} \\
&= \frac{\sum_{\mathbf{u}} p(Y \in \mathcal{B}, \mathbf{Z} = \mathbf{z} \mid \mathbf{A} = \mathbf{a}, \mathrm{Pa}_{\mathbf{A}}^{\mathcal{G}} = \mathbf{u}) p(\mathrm{Pa}_{\mathbf{A}}^{\mathcal{G}} = \mathbf{u})}{\sum_{\mathbf{u}} p(\mathbf{Z} = \mathbf{z} \mid \mathbf{A} = \mathbf{a}, \mathrm{Pa}_{\mathbf{A}}^{\mathcal{G}} = \mathbf{u}) p(\mathrm{Pa}_{\mathbf{A}}^{\mathcal{G}} = \mathbf{u})} \\
&= \frac{\sum_{\mathbf{u}} p(Y \in \mathcal{B} \mid \mathbf{Z} = \mathbf{z}, \mathbf{A} = \mathbf{a}, \mathrm{Pa}_{\mathbf{A}}^{\mathcal{G}} = \mathbf{u}) p(\mathbf{Z} = \mathbf{z} \mid \mathbf{A} = \mathbf{a}, \mathrm{Pa}_{\mathbf{A}}^{\mathcal{G}} = \mathbf{u}) p(\mathrm{Pa}_{\mathbf{A}}^{\mathcal{G}} = \mathbf{u})}{\sum_{\mathbf{u}} p(\mathbf{Z} = \mathbf{z} \mid \mathbf{A} = \mathbf{a}, \mathrm{Pa}_{\mathbf{A}}^{\mathcal{G}} = \mathbf{u}) p(\mathrm{Pa}_{\mathbf{A}}^{\mathcal{G}} = \mathbf{u})} \\
&= \frac{\sum_{\mathbf{u}} p(Y \in \mathcal{B} \mid \mathbf{Z} = \mathbf{z}) p(\mathbf{Z} = \mathbf{z} \mid \mathbf{A} = \mathbf{a}, \mathrm{Pa}_{\mathbf{A}}^{\mathcal{G}} = \mathbf{u}) p(\mathrm{Pa}_{\mathbf{A}}^{\mathcal{G}} = \mathbf{u})}{\sum_{\mathbf{u}} p(\mathbf{Z} = \mathbf{z} \mid \mathbf{A} = \mathbf{a}, \mathrm{Pa}_{\mathbf{A}}^{\mathcal{G}} = \mathbf{u}) p(\mathrm{Pa}_{\mathbf{A}}^{\mathcal{G}} = \mathbf{u})} \\
&= p(Y \in \mathcal{B} \mid \mathbf{Z} = \mathbf{z}),
\end{aligned}
$$

where the second last step uses Lemma A.9.

Conversely, suppose there exists $p \in \mathscr{P}_{\mathbf{V}}$ that is Markov relative to $\mathcal{G}$ and under which $Y \not\perp \mathbf{A} \mid \mathbf{Z}$. This implies (see the remark following Theorem 1.2.4 in [29]) there exists $p \in \mathscr{P}_{\mathbf{V}}$ with $p(Y \in \mathcal{B} \mid \mathbf{Z} = \mathbf{z}, \mathbf{A} = \mathbf{a}, \mathrm{Pa}_{\mathbf{A}}^{\mathcal{G}} = \mathbf{u}) \neq p(Y \in \mathcal{B} \mid \mathbf{Z} = \mathbf{z})$. By the above, this implies that $p_a(Y \in \mathcal{B} \mid \mathbf{Z} = \mathbf{z}) \neq p(Y \in \mathcal{B} \mid \mathbf{Z} = \mathbf{z})$. $\qquad \square$

## A.3 Proof of Theorem 3.4

If $\mathbf{Z}$ $d$-separates $Y$ from $\mathbf{A}$ on $\mathcal{G}_{\bar{\mathbf{A}}}$, then by Theorem 3.3 $\{p_a(\mathbf{Z}, Y) : a \in \mathcal{A}\}$ is conditionally benign for every $p \in \mathscr{P}_{\mathbf{V}}$ that is Markov relative to $\mathcal{G}_{\bar{\mathbf{A}}}$, and hence is still conditionally benign when the null intervention is excluded. It remains to observe that for any $p$ that is Markov relative to $\mathcal{G}$, there exists $p'$ that is Markov relative to $\mathcal{G}_{\bar{\mathbf{A}}}$ such that

$$\{p'_a(\mathbf{Z}, Y) : a \in \mathcal{A}_0\} = \{p_a(\mathbf{Z}, Y) : a \in \mathcal{A}_0\}.$$

Conversely, suppose there exists $p \in \mathscr{P}_{\mathbf{V}}$ that is Markov relative to $\mathcal{G}_{\bar{\mathbf{A}}}$ and under which $Y \not\perp \mathbf{A} \mid \mathbf{Z}$. Since necessarily $p_a(Y \in \mathcal{B} \mid \mathbf{Z} = \mathbf{z}) = p(Y \in \mathcal{B} \mid \mathbf{Z} = \mathbf{z})$ when $a$ is the null intervention, it must be some $a \in \mathcal{A}_0$ that realizes the failure from the proof of Theorem 3.3, which means $\{p_a(\mathbf{Z}, Y) : a \in \mathcal{A}_0\}$ is not conditionally benign. $\qquad \square$

## A.4 Proof of Lemma 3.5

Suppose that $\mathbf{Z}$ satisfies the front-door criterion relative to $(\mathbf{A}, Y)$ on $\mathcal{G}$ and there exists a path from $\mathbf{A}$ to $Y$ on $\mathcal{G}_{\bar{\mathbf{A}}}$ that is unblocked by $\mathbf{Z}$. The path cannot be directed, since by the front-door criterion (a) it would include the subpath $\bigcirc \to \mathbf{Z} \to \bigcirc$, and hence would be blocked. The path also cannot be a back-door path since there are no arrows going into $\mathbf{A}$ on $\mathcal{G}_{\bar{\mathbf{A}}}$. Thus, there must be some part of the path that is of the form $\bigcirc \to B \leftarrow \bigcirc$ for some variable $B \in \mathbf{V}$ and there are no remaining colliders on the subpath from $B$ to $Y$. Since the path is unblocked, this $B$ must have a descendant (potentially itself) in $\mathbf{Z}$, and this creates a back-door path from $\mathbf{Z}$ to $Y$. On the portion of the back-door path that is from $\mathbf{Z}$ to $B$, there can be no colliders since then $\mathbf{Z}$ would not be a descendant of $B$, and hence this backdoor path contains no colliders. Since the back-door path also does not contain $\mathbf{A}$, it is unblocked by $\mathbf{A}$, which violates the front-door criterion (c). Thus, no path from $\mathbf{A}$ to $Y$ on $\mathcal{G}_{\bar{\mathbf{A}}}$ that is unblocked by $\mathbf{Z}$ can exist, so $\mathbf{Z}$ blocks every path and hence $\mathbf{Z}$ $d$-separates $Y$ from $\mathbf{A}$ on $\mathcal{G}_{\bar{\mathbf{A}}}$. $\qquad\square$

## B Proofs for Regret Bounds

### B.1 Concentration of Empirical Means

For a fixed $t \in [T]$, $a \in \mathcal{A}$, and $z \in \mathcal{Z}$, define the events

$$
E_t^{\mathcal{A}}(a) = \left\{ \left| \hat{\mu}_t^{\mathcal{A}}(a) - \mathbb{E}_{\nu_a}[Y] \right| \leq \sqrt{\frac{\log(2/\delta)}{2\mathbb{T}_t^{\mathcal{A}}(a)}} \right\}
$$

and

$$
E_t^{\mathcal{Z}}(z) = \left\{ \max_{a \in \mathcal{A}} \left| \hat{\mu}_t^{\mathcal{Z}}(z) - \mathbb{E}_{\nu_a}[Y \mid Z = z] \right| \leq \sqrt{\frac{\log(2/\delta)}{2\mathbb{T}_t^{\mathcal{Z}}(z)}} \right\}.
$$

Let $E^{\mathcal{A}} = \cap_{t \in [T], a \in \mathcal{A}} E_t^{\mathcal{A}}(a)$, $E^{\mathcal{Z}} = \cap_{t \in [T], z \in \mathcal{Z}} E_t^{\mathcal{Z}}(z)$, and $E = E^{\mathcal{A}} \cap E^{\mathcal{Z}}$. Finally, define the event

$$
E^{\nu} = \left\{ \sup_{a \in \mathcal{A}} \sum_{z \in \mathcal{Z}} \left| \mathbb{P}_{\hat{\nu}_a}[Z = z] - \mathbb{P}_{\nu_a}[Z = z] \right| \leq \frac{\sqrt{|\mathcal{A}| \, |\mathcal{Z}| \log T}}{T^{1/4}} \right\}.
$$

**Lemma B.1.** *For any $\nu \in \mathscr{P}(\mathcal{Z} \times \mathcal{Y})^{\mathcal{A}}$ and $\pi \in \Pi(\mathcal{A}, \mathcal{Z}, T)$,*

$$
\mathbb{P}_{\nu,\pi}[(E^{\mathcal{A}})'] \leq |\mathcal{A}| \, T\delta.
$$

*Proof of Lemma B.1.* For each $a \in \mathcal{A}$, define the new i.i.d. random variables $Y_1^{\circ}(a), \ldots, Y_T^{\circ}(a) \sim \nu_a$. For any $t \in [T]$, Hoeffding's inequality can be applied to obtain

$$
\mathbb{P}_{\nu_a}\left( \left| \frac{1}{t} \sum_{s=1}^{t} Y_s^{\circ}(a) - \mathbb{E}_{\nu_a}[Y] \right| > \sqrt{\frac{\log(2/\delta)}{2t}} \right) \leq \delta.
$$

Then, using the i.i.d. property of $(Y_1(a), \ldots, Y_T(a))$,

$$
\mathbb{P}_{\nu,\pi}\left( \exists t \in [T], a \in \mathcal{A} : \quad \left| \hat{\mu}_t^{\mathcal{A}}(a) - \mathbb{E}_{\nu_a}[Y] \right| > \sqrt{\frac{\log(2/\delta)}{2\mathbb{T}_t^{\mathcal{A}}(a)}} \right)
$$

$$
\leq \sum_{t=1}^{T} \sum_{a \in \mathcal{A}} \mathbb{P}_{\nu_a}\left( \left| \frac{1}{t} \sum_{s=1}^{t} Y_s^{\circ}(a) - \mathbb{E}_{\nu_a}[Y] \right| > \sqrt{\frac{\log(2/\delta)}{2t}} \right)
$$

$$
\leq |\mathcal{A}| \, T\delta,
$$

where we have used a union bound over $a \in \mathcal{A}$ and $t \in [T]$. $\qquad\square$

**Lemma B.2.** *For any $\nu \in \mathscr{P}(\mathcal{Z} \times \mathcal{Y})^{\mathcal{A}}$ that is conditionally benign and $\pi$,*

$$
\mathbb{P}_{\nu,\pi}[(E^{\mathcal{Z}})'] \leq |\mathcal{Z}| \, T\delta.
$$

*Proof of Lemma B.2.* Since $\nu$ is conditionally benign, there exists $p \in \mathscr{P}(\mathcal{Z} \times \mathcal{Y})$ such that for each $a \in \mathcal{A}$, $\nu_a(Y \mid Z) = p(Y \mid Z)$. Fix $z \in \mathcal{Z}$, and define joint the distribution $q_z(Y, Z) = p(Y \mid Z)\mathbb{I}\{Z = z\}$. Finally, define the new i.i.d. random variables $V_1^\circ, \ldots, V_T^\circ \sim q_z$. For any $t \in [T]$, Hoeffding's inequality can be applied to obtain

$$\mathbb{P}_{q_z}\left( \left| \frac{1}{t} \sum_{s=1}^{t} V_s^\circ - \mathbb{E}_{q_z}[Y] \right| \leq \sqrt{\frac{\log(2/\delta)}{2t}} \right) \leq \delta.$$

Then,

$$\mathbb{P}_{\nu,\pi}\left( \exists t \in [T], z \in \mathcal{Z}: \quad \max_{a \in \mathcal{A}} \left| \hat{\mu}_t^{\mathcal{Z}}(z) - \mathbb{E}_{\nu_a}[Y \mid Z = z] \right| > \sqrt{\frac{\log(2/\delta)}{2\mathbb{T}_t^{\mathcal{Z}}(z)}} \right)$$

$$= \mathbb{P}_{\nu,\pi}\left( \exists t \in [T], z \in \mathcal{Z}: \quad \left| \hat{\mu}_t^{\mathcal{Z}}(z) - \mathbb{E}_p[Y \mid Z = z] \right| \leq \sqrt{\frac{\log(2/\delta)}{2\mathbb{T}_t^{\mathcal{Z}}(z)}} \right)$$

$$\leq \sum_{t=1}^{T} \sum_{z \in \mathcal{Z}} \mathbb{P}_{q_z}\left( \left| \frac{1}{t} \sum_{s=1}^{t} V_s^\circ - \mathbb{E}_{q_z}[Y] \right| \leq \sqrt{\frac{\log(2/\delta)}{2t}} \right)$$

$$\leq |\mathcal{Z}| T \delta.$$

where we have used a union bound over $z \in \mathcal{Z}$ and $t \in [T]$. $\square$

**Theorem B.3** (Theorem 1 of Canonne [11]). *Let $p$ be any distribution on $[k]$ for some integer $k$. For any $\varepsilon, \delta > 0$, if $n \geq \max\{k/\varepsilon^2, (2/\varepsilon^2)\log(2/\delta)\}$ and $X_1, \ldots, X_n$ is an i.i.d. sample from $p$, then the MLE estimator*

$$\hat{p}_n(j) = \frac{1}{n} \sum_{t=1}^{n} \mathbb{I}\{X_t = j\} \quad \forall j \in [k]$$

*satisfies*

$$\mathbb{P}\left[ \frac{1}{2} \sum_{j \in [k]} |\hat{p}_n(j) - p(j)| > \varepsilon \right] \leq \delta.$$

**Lemma B.4.** *If $\hat{\nu}(Z)$ is estimated using uniform exploration of at least $\lceil 4\sqrt{T}/|\mathcal{A}| \rceil$ rounds for each $a \in \mathcal{A}$,*

$$\mathbb{P}_{\nu,\pi}[(E^\nu)'] \leq 2|\mathcal{A}|/T.$$

*Proof.* Let $\varepsilon = (1/2)T^{-1/4}\sqrt{|\mathcal{A}| |\mathcal{Z}| \log T}$, $\delta = 2/T$, and $n$ denote the number of exploration rounds used to estimate each $\hat{\nu}_a$. By a union bound and Theorem B.3, if $n \geq \max\{|\mathcal{Z}|/\varepsilon^2, (2/\varepsilon^2)\log(2/\delta)\}$ then

$$\mathbb{P}_{\nu,\pi}[(E^\nu)'] \leq \mathbb{P}_{\nu,\pi}\left[ \exists a \in \mathcal{A}: \sum_{z \in \mathcal{Z}} \left| \mathbb{P}_{\hat{\nu}_a}[Z = z] - \mathbb{P}_{\nu_a}[Z = z] \right| > 2\varepsilon \right]$$

$$\leq \sum_{a \in \mathcal{A}} \mathbb{P}_{\nu,\pi}\left[ \frac{1}{2} \sum_{z \in \mathcal{Z}} \left| \mathbb{P}_{\hat{\nu}_a}[Z = z] - \mathbb{P}_{\nu_a}[Z = z] \right| > \varepsilon \right]$$

$$\leq 2|\mathcal{A}|/T.$$

Then, it remains to observe that when $T \geq 3$ and $|\mathcal{Z}| \geq 2$ (which can be trivially assumed, since $\hat{\nu}$ is known exactly if $|\mathcal{Z}| = 1$),

$$\frac{|\mathcal{Z}|}{\varepsilon^2} = \frac{4|\mathcal{Z}|\sqrt{T}}{|\mathcal{A}| |\mathcal{Z}| \log T} \leq \frac{4\sqrt{T}}{|\mathcal{A}|}$$

and

$$\frac{2}{\varepsilon^2} \log(2/\delta) = \frac{8\sqrt{T}}{|\mathcal{A}| |\mathcal{Z}| \log T}(\log T) \leq \frac{4\sqrt{T}}{|\mathcal{A}|}.$$

$\square$

## B.2 Bounding Accumulated Regret

**Lemma B.5.** *For any $\nu \in \mathscr{P}(\mathcal{Z} \times \mathcal{Y})^{\mathcal{A}}$, $\pi \in \Pi(\mathcal{A}, \mathcal{Z}, T)$, and $t < t' \in [T]$, it holds almost surely that*

$$\sum_{s=t}^{t'} \frac{1}{\sqrt{\mathbb{T}_{s-1}^{\mathcal{A}}(A_s)}} \leq \sqrt{8\,|\mathcal{A}|\,(t'-t)}\,.$$

*Proof of Lemma B.5.* Using Lemma 4.13 of Orabona [28],

$$\sum_{s=t}^{t'} \frac{1}{\sqrt{\mathbb{T}_{s-1}^{\mathcal{A}}(A_s)}} = \sum_{s=t}^{t'} \sum_{a \in \mathcal{A}} \frac{\mathbb{I}\{A_s = a\}}{\sqrt{1 \vee \sum_{j=1}^{s-1} \mathbb{I}\{A_j = a\}}}$$

$$\leq \sum_{s=t}^{t'} \sum_{a \in \mathcal{A}} \frac{\mathbb{I}\{A_s = a\}}{\sqrt{\sum_{j=1}^{t-1} \mathbb{I}\{A_j = a\} + (1/2)\sum_{j=t}^{s} \mathbb{I}\{A_j = a\}}}$$

$$\leq \sqrt{2} \sum_{a \in \mathcal{A}} \int_{\sum_{j=1}^{t-1} \mathbb{I}\{A_j=a\}}^{\sum_{j=1}^{t'} \mathbb{I}\{A_j=a\}} x^{-1/2} \mathrm{d}x$$

$$= \sqrt{8} \sum_{a \in \mathcal{A}} \left( \sqrt{\sum_{j=1}^{t'} \mathbb{I}\{A_j = a\}} - \sqrt{\sum_{j=1}^{t-1} \mathbb{I}\{A_j = a\}} \right)$$

$$\leq \sum_{a \in \mathcal{A}} \sqrt{8 \sum_{j=t}^{t'} \mathbb{I}\{A_j = a\}}$$

$$\leq |\mathcal{A}| \sqrt{\frac{8}{|\mathcal{A}|} \sum_{a \in \mathcal{A}} \sum_{j=t}^{t'} \mathbb{I}\{A_j = a\}}$$

$$= \sqrt{8\,|\mathcal{A}|\,(t'-t)}.$$

$\square$

**Lemma B.6.** *For any $\nu \in \mathscr{P}(\mathcal{Z} \times \mathcal{Y})^{\mathcal{A}}$, $\pi \in \Pi(\mathcal{A}, \mathcal{Z}, T)$, and $t < t' \in [T]$,*

$$\mathbb{E}_{\nu,\pi} \left[ \sum_{s=t}^{t'} \sum_{z \in \mathcal{Z}} \frac{1}{\sqrt{\mathbb{T}_{s-1}^{\mathcal{Z}}(z)}} \mathbb{P}_{\nu_{A_s}}[Z = z] \right] \leq \sqrt{8\,|\mathcal{Z}|\,(t'-t)} + \sqrt{(1/2)(t'-t)\log(t'-t)} + 2.$$

*Proof of Lemma B.6.* First,

$$\mathbb{E}_{\nu,\pi} \left[ \sum_{s=t}^{t'} \sum_{z \in \mathcal{Z}} \sqrt{1/\mathbb{T}_{s-1}^{\mathcal{Z}}(z)}\, \mathbb{P}_{\nu_{A_s}}[Z = z] \right]$$

$$= \mathbb{E}_{\nu,\pi} \left[ \sum_{s=t}^{t'} \sum_{z \in \mathcal{Z}} \sqrt{1/\mathbb{T}_{s-1}^{\mathcal{Z}}(z)}\, \mathbb{I}\{Z_s(A_s) = z\} \right]$$

$$+ \mathbb{E}_{\nu,\pi} \left[ \sum_{s=t}^{t'} \sum_{z \in \mathcal{Z}} \sqrt{1/\mathbb{T}_{s-1}^{\mathcal{Z}}(z)} \left( \mathbb{P}_{\nu_{A_s}}[Z = z] - \mathbb{I}\{Z_s(A_s) = z\} \right) \right]$$

$$\leq \sqrt{8\,|\mathcal{Z}|\,(t'-t)} + \mathbb{E}_{\nu,\pi} \left[ \sum_{s=t}^{t'} \sum_{z \in \mathcal{Z}} \sqrt{1/\mathbb{T}_{s-1}^{\mathcal{Z}}(z)} \left( \mathbb{P}_{\nu_{A_s}}[Z = z] - \mathbb{I}\{Z_s(A_s) = z\} \right) \right],$$

where we have used the same argument as Lemma B.5 applied to $\mathbb{T}_{s-1}^{\mathcal{Z}}(z)$ rather than $\mathbb{T}_{s-1}^{\mathcal{A}}(a)$. Following the analysis of Lu et al. [24], for $t \leq j \leq t'$ define the random variable

$$M_j = \sum_{s=t}^{j} \sum_{z \in \mathcal{Z}} \sqrt{1/\mathbb{T}_{s-1}^{\mathcal{Z}}(z)} \left( \mathbb{P}_{\nu_{A_s}}[Z = z] - \mathbb{I}\{Z_s(A_s) = z\} \right).$$

Then,

$$\mathbb{E}_{\nu,\pi}[M_j \mid A_j, H_{j-1}] = M_{j-1} + \sum_{z \in \mathcal{Z}} \sqrt{1/\mathbb{T}_{j-1}^{\mathcal{Z}}(z)}\, \mathbb{E}_{\nu,\pi}\left[\mathbb{P}_{\nu_{A_s}}[Z=z] - \mathbb{I}\{Z_j(A_j)=z\} \mid A_j\right] = M_{j-1}.$$

Further, it holds almost surely that

$$
\begin{aligned}
|M_j - M_{j-1}| &= \left| \sum_{z \in \mathcal{Z}} \sqrt{1/\mathbb{T}_{j-1}^{\mathcal{Z}}(z)}\left[\mathbb{P}_{\nu_{A_s}}[Z=z] - \mathbb{I}\{Z_j(A_j)=z\}\right]\right| \\
&= \left| \mathbb{E}_{\nu,\pi}\left[ \sum_{z \in \mathcal{Z}} \sqrt{1/\mathbb{T}_{j-1}^{\mathcal{Z}}(z)}\,\mathbb{I}\{Z_j(A_j)=z\} \,\Big|\, A_j, H_{j-1}\right] - \sum_{z \in \mathcal{Z}} \sqrt{1/\mathbb{T}_{j-1}^{\mathcal{Z}}(z)}\,\mathbb{I}\{Z_j(A_j)=z\}\right| \\
&= \left| \mathbb{E}_{\nu,\pi}\left[ \sqrt{1/\mathbb{T}_{j-1}^{\mathcal{Z}}(Z_j(A_j))} \mid A_j, H_{j-1}\right] - \sqrt{1/\mathbb{T}_{j-1}^{\mathcal{Z}}(Z_j(A_j))}\right| \\
&\leq 1.
\end{aligned}
$$

Then, by Azuma-Hoeffding, for all $x > 0$

$$\mathbb{P}_{\nu,\pi}\left[ |M_{t'}| > \sqrt{x(t'-t)\log(t'-t)}\right] \leq 2e^{-2x\log(t'-t)}.$$

Thus, since $|M_{t'}| \leq t' - t$,

$$
\begin{aligned}
\mathbb{E}_{\nu,\pi}&\left[ \sum_{s=t}^{t'} \sum_{z \in \mathcal{Z}} \sqrt{1/\mathbb{T}_{t-1}^{\mathcal{Z}}(z)}\left(\mathbb{P}_{\nu_{A_s}}[Z=z] - \mathbb{I}\{Z_s(A_s)=z\}\right)\right] \\
&\leq 2(t'-t)e^{-2x\log(t'-t)} + \sqrt{x(t'-t)\log(t'-t)}.
\end{aligned}
$$

Taking $x = 1/2$ gives the result. $\qquad\square$

### B.3   Proof of Theorem 4.3

First, by Lemma B.2

$$
\begin{aligned}
R_{\nu,\mathrm{C}}(T) &= \mathbb{E}_{\nu,\mathrm{C}} \sum_{t=1}^{T}\left[\mathbb{E}_{\nu_{a_\nu^*}}[Y] - \mathbb{E}_{\nu_{A_t^{\mathrm{C}}}}[Y]\right] \\
&\leq |\mathcal{Z}|\, T^2 \delta + \mathbb{E}_{\nu,\mathrm{C}}\left[\mathbb{I}\{E^{\mathcal{Z}}\} \sum_{t=1}^{T}\left(\mathbb{E}_{\nu_{a_\nu^*}}[Y] - \mathbb{E}_{\nu_{A_t^{\mathrm{C}}}}[Y]\right)\right].
\end{aligned}
\tag{B.1}
$$

Then, by the conditionally benign property,

$$
\begin{aligned}
\mathbb{E}_{\nu,\mathrm{C}}&\left[\mathbb{I}\{E^{\mathcal{Z}}\} \sum_{t=1}^{T}\left(\mathbb{E}_{\nu_{a_\nu^*}}[Y] - \mathbb{E}_{\nu_{A_t^{\mathrm{C}}}}[Y]\right)\right] \\
&= \mathbb{E}_{\nu,\mathrm{C}}\left[\mathbb{I}\{E^{\mathcal{Z}}\} \sum_{t=1}^{T}\left(\mathbb{E}_{\nu_{a_\nu^*}}[Y] - \widetilde{\mathrm{UCB}}_t(A_t^{\mathrm{C}}) + \widetilde{\mathrm{UCB}}_t(A_t^{\mathrm{C}}) - \mathbb{E}_{\nu_{A_t^{\mathrm{C}}}}[Y]\right)\right] \\
&= \mathbb{E}_{\nu,\mathrm{C}}\left[\mathbb{I}\{E^{\mathcal{Z}}\} \sum_{t=1}^{T}\left(\sum_{z \in \mathcal{Z}} \mathbb{E}_{\nu_{a_\nu^*}}[Y \mid Z=z]\mathbb{P}_{\nu_{a_\nu^*}}[Z=z] - \widetilde{\mathrm{UCB}}_t(A_t^{\mathrm{C}})\right)\right] \\
&\quad + \mathbb{E}_{\nu,\mathrm{C}}\left[\mathbb{I}\{E^{\mathcal{Z}}\} \sum_{t=1}^{T}\left(\widetilde{\mathrm{UCB}}_t(A_t^{\mathrm{C}}) - \sum_{z \in \mathcal{Z}} \mathbb{E}_{\nu_{A_t^{\mathrm{C}}}}[Y \mid Z=z]\mathbb{P}_{\nu_{A_t^{\mathrm{C}}}}[Z=z]\right)\right].
\end{aligned}
$$

We bound these terms separately. First, using the fact that $\tilde{\nu}(Z)$ and $\nu(Z)$ are $\varepsilon$-close, the definition of $E^{\mathcal{Z}}$, and the definition of $A_t^{\mathrm{C}}$,

$$\mathbb{E}_{\nu,\mathrm{C}}\Big[\mathbb{I}\{E^{\mathcal{Z}}\}\sum_{t=1}^{T}\Big(\sum_{z\in\mathcal{Z}}\mathbb{E}_{\nu_{a_\nu^*}}[Y\,|\,Z=z]\mathbb{P}_{\nu_{a_\nu^*}}[Z=z]-\widetilde{\mathrm{UCB}}_t(A_t^{\mathrm{C}})\Big)\Big]$$

$$\leq \mathbb{E}_{\nu,\mathrm{C}}\Big[\mathbb{I}\{E^{\mathcal{Z}}\}\sum_{t=1}^{T}\Big(\sum_{z\in\mathcal{Z}}\mathbb{E}_{\nu_{a_\nu^*}}[Y\,|\,Z=z]\mathbb{P}_{\tilde{\nu}_{a_\nu^*}}[Z=z]-\widetilde{\mathrm{UCB}}_t(A_t^{\mathrm{C}})\Big)\Big]+\varepsilon T \qquad (\text{B.2})$$

$$\leq \mathbb{E}_{\nu,\mathrm{C}}\Big[\mathbb{I}\{E^{\mathcal{Z}}\}\sum_{t=1}^{T}\Big(\widetilde{\mathrm{UCB}}_t(a_\nu^*)-\widetilde{\mathrm{UCB}}_t(A_t^{\mathrm{C}})\Big)\Big]+\varepsilon T$$

$$\leq \varepsilon T.$$

Second, using the definition of $E^{\mathcal{Z}}$, the fact that $\tilde{\nu}(Z)$ and $\nu(Z)$ are $\varepsilon$-close, the conditionally benign property, and Lemma B.6,

$$\mathbb{E}_{\nu,\mathrm{C}}\Big[\mathbb{I}\{E^{\mathcal{Z}}\}\sum_{t=1}^{T}\Big(\widetilde{\mathrm{UCB}}_t(A_t^{\mathrm{C}})-\mathbb{E}_{\nu_{A_t^{\mathrm{C}}}}[Y]\Big)\Big]$$

$$=\mathbb{E}_{\nu,\mathrm{C}}\Big[\mathbb{I}\{E^{\mathcal{Z}}\}\sum_{t=1}^{T}\Big(\sum_{z\in\mathcal{Z}}\Big(\hat{\mu}_t^{\mathcal{Z}}(z)+\sqrt{\log(2/\delta)/(2\mathbb{T}_t^{\mathcal{Z}}(z))}\Big)\mathbb{P}_{\tilde{\nu}_{A_t^{\mathrm{C}}}}[Z=z]-\mathbb{E}_{\nu_{A_t^{\mathrm{C}}}}[Y]\Big)\Big]$$

$$\leq\mathbb{E}_{\nu,\mathrm{C}}\Big[\mathbb{I}\{E^{\mathcal{Z}}\}\sum_{t=1}^{T}\Big(\sum_{z\in\mathcal{Z}}\Big(\mathbb{E}_{\nu_{A_t^{\mathrm{C}}}}[Y\,|\,Z=z]+\sqrt{2\log(2/\delta)/\mathbb{T}_t^{\mathcal{Z}}(z)}\Big)\mathbb{P}_{\tilde{\nu}_{A_t^{\mathrm{C}}}}[Z=z]-\mathbb{E}_{\nu_{A_t^{\mathrm{C}}}}[Y]\Big)\Big]$$

$$\leq\mathbb{E}_{\nu,\mathrm{C}}\Big[\mathbb{I}\{E^{\mathcal{Z}}\}\sum_{t=1}^{T}\Big(\sum_{z\in\mathcal{Z}}\Big(\mathbb{E}_{\nu_{A_t^{\mathrm{C}}}}[Y\,|\,Z=z]+\sqrt{2\log(2/\delta)/\mathbb{T}_t^{\mathcal{Z}}(z)}\Big)\mathbb{P}_{\nu_{A_t^{\mathrm{C}}}}[Z=z]-\mathbb{E}_{\nu_{A_t^{\mathrm{C}}}}[Y]\Big)\Big]$$

$$\quad+\varepsilon\Big(1+\sqrt{2\log(2/\delta)}\Big)T$$

$$=\mathbb{E}_{\nu,\mathrm{C}}\Big[\mathbb{I}\{E^{\mathcal{Z}}\}\sum_{t=1}^{T}\sum_{z\in\mathcal{Z}}\sqrt{2\log(2/\delta)/\mathbb{T}_t^{\mathcal{Z}}(z)}\,\mathbb{P}_{\nu_{A_t^{\mathrm{C}}}}[Z=z]\Big]+\varepsilon\Big(1+\sqrt{2\log(2/\delta)}\Big)T$$

$$\leq 4\sqrt{2\,|\mathcal{Z}|\,T\log T}+(\log T)\sqrt{2T}+4\sqrt{\log T}+\varepsilon(1+2\sqrt{\log T})T.$$

$$(\text{B.3})$$

where the last line follows by taking $\delta=2/T^2$. The theorem then follows by combining Eqs. (B.1) to (B.3). $\qquad\square$

## B.4 Proof of Theorem 4.5

We may assume without loss of generality that

$$\lim_{T\to\infty}\frac{\log(1/\delta_T)}{T}=0.$$

If this is not the case, then for large enough $T$, it holds that $\log(1/\delta_T)\geq cT$ for some $c$, and hence the instance dependent lower bounds for UCB variants [6] imply that regret grows linearly in $T$ in the worst-case. We may also assume $|\mathcal{Z}|>1$, for otherwise C-UCB plays the same arm forever (using its arbitrary tie-break rule) and so C-UCB can be forced to incur linear regret in a trivial way.

To illustrate that the lower bound is witnessed by a diversity of environments, we describe the construction in more general terms and then provide an example instantiation at the end of the proof. Let $\mathcal{A}_0$ and $\mathcal{Z}_0$ be arbitrary, nonempty, strict subsets of $\mathcal{A}$ and $\mathcal{Z}$ respectively, and let $\mathcal{A}_1=\mathcal{A}\setminus\mathcal{A}_0$ and $\mathcal{Z}_1=\mathcal{Z}\setminus\mathcal{Z}_0$.

We now describe sufficient conditions for an environment to be not conditionally benign and to force C-UCB to incur linear regret. For $p^{\mathcal{A}}(0)$ and $p^{\mathcal{A}}(1)$ in $(0,1)$, let the marginal distribution be such

that for $i \in \{0, 1\}$, if $a \in \mathcal{A}_i$ then

$$\mathbb{P}_{\nu_a}[Z = z] = \frac{p^{\mathcal{A}}(i)}{|\mathcal{Z}_1|}\mathbb{I}\{z \in \mathcal{Z}_1\} + \frac{1 - p^{\mathcal{A}}(i)}{|\mathcal{Z}_0|}\mathbb{I}\{z \in \mathcal{Z}_0\}.$$

Similarly, for $\mu^{\mathcal{A},\mathcal{Z}}(0,0)$, $\mu^{\mathcal{A},\mathcal{Z}}(0,1)$, $\mu^{\mathcal{A},\mathcal{Z}}(1,0)$, and $\mu^{\mathcal{A},\mathcal{Z}}(1,1)$ in $(0, 1)$, let the conditional distribution be such that for $i, j \in \{0, 1\}$, if $a \in \mathcal{A}_i$ and $z \in \mathcal{Z}_j$ then

$$\nu_a(Y \mid Z = z) = \text{Ber}(\mu^{\mathcal{A},\mathcal{Z}}(i, j)).$$

Observe that for $i \in \{0, 1\}$, if $a \in \mathcal{A}_i$ then

$$\mathbb{E}_{\nu_a}[Y] = \mu^{\mathcal{A},\mathcal{Z}}(i, 1)p^{\mathcal{A}}(i) + \mu^{\mathcal{A},\mathcal{Z}}(i, 0)[1 - p^{\mathcal{A}}(i)].$$

We suppose the following conditions:

(1) $\forall a \in \mathcal{A}_0, a' \in \mathcal{A}_1 \quad \mathbb{E}_{\nu_a}[Y] > \mathbb{E}_{\nu_{a'}}[Y]$,

(2) $p^{\mathcal{A}}(0) < p^{\mathcal{A}}(1)$,

(3) $\min\{\mu^{\mathcal{A},\mathcal{Z}}(0,1), \mu^{\mathcal{A},\mathcal{Z}}(1,1)\} > \max\{\mu^{\mathcal{A},\mathcal{Z}}(0,0), \mu^{\mathcal{A},\mathcal{Z}}(1,0)\}$.

By Condition (1), $a_\nu^* \in \mathcal{A}_0$ (note that then all $a \in \mathcal{A}_0$ are equally optimal), so a constant amount of regret is incurred whenever $A_t^C \in \mathcal{A}_1$. We now argue that under Conditions (2) and (3), this happens for a constant fraction of rounds with high probability.

First, we require slightly more notation to understand the behaviour of $A_t^C$. For every $t \in [T]$, $i \in \{0, 1\}$, and $z \in \mathcal{Z}$, let

$$\mathbb{T}_t^{\mathcal{A},\mathcal{Z}}(i, z) = \sum_{s=1}^{t} \mathbb{I}\{A_s^C \in \mathcal{A}_i, Z_s = z\}.$$

Define $\mathbb{T}_t^{\mathcal{A}}(i) = \sum_{z \in \mathcal{Z}} \mathbb{T}_t^{\mathcal{A},\mathcal{Z}}(i, z)$, and note that $\mathbb{T}_t^{\mathcal{Z}}(z) = \mathbb{T}_t^{\mathcal{A},\mathcal{Z}}(0, z) + \mathbb{T}_t^{\mathcal{A},\mathcal{Z}}(1, z)$. Further, let

$$\hat{\mu}_t^{\mathcal{A},\mathcal{Z}}(i, z) = \frac{1}{\mathbb{T}_t^{\mathcal{A},\mathcal{Z}}(i, z)}\sum_{s=1}^{t} Y_s(A_s^C)\mathbb{I}\{A_s^C \in \mathcal{A}_i, Z_s = z\},$$

By definition,

$$\hat{\mu}_t^{\mathcal{Z}}(z) = \frac{\mathbb{T}_t^{\mathcal{A},\mathcal{Z}}(0, z)}{\mathbb{T}_t^{\mathcal{Z}}(z)}\hat{\mu}_t^{\mathcal{A},\mathcal{Z}}(0, z) + \frac{\mathbb{T}_t^{\mathcal{A},\mathcal{Z}}(1, z)}{\mathbb{T}_t^{\mathcal{Z}}(z)}\hat{\mu}_t^{\mathcal{A},\mathcal{Z}}(1, z).$$

Define the event $F$ to be the case that for all $t \in [T]$, $i, j \in \{0, 1\}$, and $z \in \mathcal{Z}_j$,

$$\left|\hat{\mu}_t^{\mathcal{A},\mathcal{Z}}(i, z) - \mu^{\mathcal{A},\mathcal{Z}}(i, j)\right| \leq \sqrt{\frac{\log T}{\mathbb{T}_t^{\mathcal{A},\mathcal{Z}}(i, z)}},$$

and let $G$ be the event that for all $t \in [T]$ and $z \in \mathcal{Z}$,

$$\frac{\mathbb{T}_t^{\mathcal{Z}}(z)}{t} \geq \min_{a \in \mathcal{A}} \mathbb{P}_{\nu_a}[Z = z] - \sqrt{\frac{\log T}{t}}$$

and

$$\frac{\mathbb{T}_t^{\mathcal{Z}}(z)}{t} \leq \max_{a \in \mathcal{A}} \mathbb{P}_{\nu_a}[Z = z] + \sqrt{\frac{\log T}{t}}.$$

By Hoeffding and a union bound (i.e., the same arguments as Lemmas B.1 and B.2), $\mathbb{P}_\nu([F \cap G]') \leq 8|\mathcal{Z}|/T$.

Now, suppose the event $F \cap G$ holds, and consider a fixed $t$. Recall that $A_t^C \in \mathcal{A}_1$ is implied by

$$\max_{a_1 \in \mathcal{A}_1} \left\{ \sum_{z \in \mathcal{Z}} \left( \hat{\mu}_t^{\mathcal{Z}}(z) + \sqrt{\frac{\log(2/\delta_T)}{2\mathbb{T}_t^{\mathcal{Z}}(z)}} \right) \mathbb{P}_{\nu_{a_1}}[Z = z] \right\}$$

$$> \max_{a_0 \in \mathcal{A}_0} \left\{ \sum_{z \in \mathcal{Z}} \left( \hat{\mu}_t^{\mathcal{Z}}(z) + \sqrt{\frac{\log(2/\delta_T)}{2\mathbb{T}_t^{\mathcal{Z}}(z)}} \right) \mathbb{P}_{\nu_{a_0}}[Z = z] \right\}.$$

This is equivalent to

$$0 < \sum_{z \in \mathcal{Z}_1} \left( \frac{\mathbb{T}_t^{\mathcal{A},\mathcal{Z}}(0,z)}{\mathbb{T}_t^{\mathcal{Z}}(z)} \hat{\mu}_t^{\mathcal{A},\mathcal{Z}}(0,z) + \frac{\mathbb{T}_t^{\mathcal{A},\mathcal{Z}}(1,z)}{\mathbb{T}_t^{\mathcal{Z}}(z)} \hat{\mu}_t^{\mathcal{A},\mathcal{Z}}(1,z) + \sqrt{\frac{\log(2/\delta)}{2\mathbb{T}_t^{\mathcal{Z}}(z)}} \right) \frac{p^{\mathcal{A}}(1) - p^{\mathcal{A}}(0)}{|\mathcal{Z}_1|}$$

$$- \sum_{z \in \mathcal{Z}_0} \left( \frac{\mathbb{T}_t^{\mathcal{A},\mathcal{Z}}(0,z)}{\mathbb{T}_t^{\mathcal{Z}}(z)} \hat{\mu}_t^{\mathcal{A},\mathcal{Z}}(0,z) + \frac{\mathbb{T}_t^{\mathcal{A},\mathcal{Z}}(1,z)}{\mathbb{T}_t^{\mathcal{Z}}(z)} \hat{\mu}_t^{\mathcal{A},\mathcal{Z}}(1,z) + \sqrt{\frac{\log(2/\delta)}{2\mathbb{T}_t^{\mathcal{Z}}(z)}} \right) \frac{p^{\mathcal{A}}(1) - p^{\mathcal{A}}(0)}{|\mathcal{Z}_0|}.$$

The following two cases hold.

(i) For all $z \in \mathcal{Z}_1$,

$$\frac{\mathbb{T}_t^{\mathcal{A},\mathcal{Z}}(0,z)}{\mathbb{T}_t^{\mathcal{Z}}(z)} \hat{\mu}_t^{\mathcal{A},\mathcal{Z}}(0,z) + \frac{\mathbb{T}_t^{\mathcal{A},\mathcal{Z}}(1,z)}{\mathbb{T}_t^{\mathcal{Z}}(z)} \hat{\mu}_t^{\mathcal{A},\mathcal{Z}}(1,z) + \sqrt{\frac{\log(2/\delta)}{2\mathbb{T}_t^{\mathcal{Z}}(z)}}$$

$$\geq \frac{\mathbb{T}_t^{\mathcal{A},\mathcal{Z}}(0,z)}{\mathbb{T}_t^{\mathcal{Z}}(z)} \left( \mu^{\mathcal{A},\mathcal{Z}}(0,1) - \sqrt{\frac{\log T}{\mathbb{T}_t^{\mathcal{A},\mathcal{Z}}(0,z)}} \right) + \frac{\mathbb{T}_t^{\mathcal{A},\mathcal{Z}}(1,z)}{\mathbb{T}_t^{\mathcal{Z}}(z)} \left( \mu^{\mathcal{A},\mathcal{Z}}(1,1) - \sqrt{\frac{\log T}{\mathbb{T}_t^{\mathcal{A},\mathcal{Z}}(1,z)}} \right)$$

$$\geq \min\{\mu^{\mathcal{A},\mathcal{Z}}(0,1), \mu^{\mathcal{A},\mathcal{Z}}(1,1)\} - 2\sqrt{\log T}\left( t\, p^{\mathcal{A}}(0)/|\mathcal{Z}_1| - \sqrt{t \log T} \right)^{-1/2}.$$

(ii) For all $z \in \mathcal{Z}_0$,

$$\frac{\mathbb{T}_t^{\mathcal{A},\mathcal{Z}}(0,z)}{\mathbb{T}_t^{\mathcal{Z}}(z)} \hat{\mu}_t^{\mathcal{A},\mathcal{Z}}(0,z) + \frac{\mathbb{T}_t^{\mathcal{A},\mathcal{Z}}(1,z)}{\mathbb{T}_t^{\mathcal{Z}}(z)} \hat{\mu}_t^{\mathcal{A},\mathcal{Z}}(1,z) + \sqrt{\frac{\log(2/\delta)}{2\mathbb{T}_t^{\mathcal{Z}}(z)}}$$

$$\leq \frac{\mathbb{T}_t^{\mathcal{A},\mathcal{Z}}(0,z)}{\mathbb{T}_t^{\mathcal{Z}}(z)} \left( \mu^{\mathcal{A},\mathcal{Z}}(0,0) + \sqrt{\frac{\log T}{\mathbb{T}_t^{\mathcal{A},\mathcal{Z}}(0,z)}} \right) + \frac{\mathbb{T}_t^{\mathcal{A},\mathcal{Z}}(1,z)}{\mathbb{T}_t^{\mathcal{Z}}(z)} \left( \mu^{\mathcal{A},\mathcal{Z}}(1,0) + \sqrt{\frac{\log T}{\mathbb{T}_t^{\mathcal{A},\mathcal{Z}}(1,z)}} \right) + \sqrt{\frac{\log(2/\delta)}{2\mathbb{T}_t^{\mathcal{Z}}(z)}}$$

$$\leq \max\{\mu^{\mathcal{A},\mathcal{Z}}(0,0), \mu^{\mathcal{A},\mathcal{Z}}(1,0)\} + 2\sqrt{\log T}\left( t(1 - p^{\mathcal{A}}(1))/|\mathcal{Z}_0| - \sqrt{t \log T} \right)^{-1/2}$$

$$+ \sqrt{\log(2/\delta)}\left( t(1 - p^{\mathcal{A}}(1))/|\mathcal{Z}_0| - \sqrt{t \log T} \right)^{-1/2}.$$

That is,

$$\sum_{z \in \mathcal{Z}_1} \left( \frac{\mathbb{T}_t^{\mathcal{A},\mathcal{Z}}(0,z)}{\mathbb{T}_t^{\mathcal{Z}}(z)} \hat{\mu}_t^{\mathcal{A},\mathcal{Z}}(0,z) + \frac{\mathbb{T}_t^{\mathcal{A},\mathcal{Z}}(1,z)}{\mathbb{T}_t^{\mathcal{Z}}(z)} \hat{\mu}_t^{\mathcal{A},\mathcal{Z}}(1,z) + \sqrt{\frac{\log(2/\delta)}{2\mathbb{T}_t^{\mathcal{Z}}(z)}} \right) \frac{p^{\mathcal{A}}(1) - p^{\mathcal{A}}(0)}{|\mathcal{Z}_1|}$$

$$- \sum_{z \in \mathcal{Z}_0} \left( \frac{\mathbb{T}_t^{\mathcal{A},\mathcal{Z}}(0,z)}{\mathbb{T}_t^{\mathcal{Z}}(z)} \hat{\mu}_t^{\mathcal{A},\mathcal{Z}}(0,z) + \frac{\mathbb{T}_t^{\mathcal{A},\mathcal{Z}}(1,z)}{\mathbb{T}_t^{\mathcal{Z}}(z)} \hat{\mu}_t^{\mathcal{A},\mathcal{Z}}(1,z) + \sqrt{\frac{\log(2/\delta)}{2\mathbb{T}_t^{\mathcal{Z}}(z)}} \right) \frac{p^{\mathcal{A}}(1) - p^{\mathcal{A}}(0)}{|\mathcal{Z}_0|}$$

$$\geq \left[ \min\{\mu^{\mathcal{A},\mathcal{Z}}(0,1), \mu^{\mathcal{A},\mathcal{Z}}(1,1)\} - 2\sqrt{\log T}\left( t\, p^{\mathcal{A}}(0)/|\mathcal{Z}_1| - \sqrt{t \log T} \right)^{-1/2} \right.$$

$$- \max\{\mu^{\mathcal{A},\mathcal{Z}}(0,0), \mu^{\mathcal{A},\mathcal{Z}}(1,0)\} - 2\sqrt{\log T}\left( t(1 - p^{\mathcal{A}}(1))/|\mathcal{Z}_0| - \sqrt{t \log T} \right)^{-1/2}$$

$$\left. - \sqrt{\log(2/\delta)}\left( t(1 - p^{\mathcal{A}}(1))/|\mathcal{Z}_0| - \sqrt{t \log T} \right)^{-1/2} \right] \left( p^{\mathcal{A}}(1) - p^{\mathcal{A}}(0) \right).$$

By Condition (3), for large enough $T$ and $t \geq T/2$, this last step can be further lower bounded[3] using

$$\min\{\mu^{\mathcal{A},\mathcal{Z}}(0,1), \mu^{\mathcal{A},\mathcal{Z}}(1,1)\} - 2\sqrt{\log T}\left( t\, p^{\mathcal{A}}(0)/|\mathcal{Z}_1| - \sqrt{t \log T} \right)^{-1/2}$$

$$- \max\{\mu^{\mathcal{A},\mathcal{Z}}(0,0), \mu^{\mathcal{A},\mathcal{Z}}(1,0)\} - 2\sqrt{\log T}\left( t(1 - p^{\mathcal{A}}(1))/|\mathcal{Z}_0| - \sqrt{t \log T} \right)^{-1/2}$$

$$- \sqrt{\log(2/\delta)}\left( t(1 - p^{\mathcal{A}}(1))/|\mathcal{Z}_0| - \sqrt{t \log T} \right)^{-1/2}$$

$$\geq \left( \min\{\mu^{\mathcal{A},\mathcal{Z}}(0,1), \mu^{\mathcal{A},\mathcal{Z}}(1,1)\} - \max\{\mu^{\mathcal{A},\mathcal{Z}}(0,0), \mu^{\mathcal{A},\mathcal{Z}}(1,0)\} \right)/2.$$

---

[3]When $\delta_T$ is polynomial in $T$, this can be improved to only require $t \geq (\log T)^2$.

Thus, we have that for sufficiently large $T$ and any $a_0 \in \mathcal{A}_0$ and $a_1 \in \mathcal{A}_1$

$$
\begin{aligned}
R_{\nu,\mathrm{C}}(T) &= \mathbb{E}_{\nu,\mathrm{C}} \sum_{t=1}^{T} (\mathbb{E}_{\nu_{a_0}}[Y] - \mathbb{E}_{\nu_{A_t^\mathrm{C}}}[Y]) \\
&\geq \mathbb{E}_{\nu,\mathrm{C}} \left[ \mathbb{I}\{F \cap G\} \sum_{t=1}^{T} (\mathbb{E}_{\nu_{a_0}}[Y] - \mathbb{E}_{\nu_{A_t^\mathrm{C}}}[Y]) \right] \\
&\geq (T/4)\Big(\mathbb{E}_{\nu_{a_0}}[Y] - \mathbb{E}_{\nu_{a_1}}[Y]\Big)(1 - 8\,|\mathcal{Z}|\,/T) \\
&\geq (T/5)\Big(\mathbb{E}_{\nu_{a_0}}[Y] - \mathbb{E}_{\nu_{a_1}}[Y]\Big).
\end{aligned}
$$

Finally, a concrete example of an environment that satisfies Conditions (1)–(3) is:

$$
\begin{aligned}
\mu^{\mathcal{A},\mathcal{Z}}(0,0) &= 1/6 & \mu^{\mathcal{A},\mathcal{Z}}(1,0) &= 2/6 \\
\mu^{\mathcal{A},\mathcal{Z}}(0,1) &= 5/6 & \mu^{\mathcal{A},\mathcal{Z}}(1,1) &= 4/6 \\
p^{\mathcal{A}}(0) &= 6/8 & p^{\mathcal{A}}(1) &= 7/8.
\end{aligned}
$$

For simplicity, we use the constants from this example in the theorem statement. $\qquad\square$

## B.5 Proof of Theorem 4.7

First, by Lemmas B.1 and B.4 with $\delta = 2/T^2$,

$$
\begin{aligned}
R_{\nu,\mathrm{HAC}}(T) &= \mathbb{E}_{\nu,\mathrm{HAC}} \sum_{t=1}^{T} \left[ \mathbb{E}_{\nu_{a_\nu^*}}[Y] - \mathbb{E}_{\nu_{A_t^\mathrm{HAC}}}[Y] \right] \\
&\leq 4\,|\mathcal{A}| + \mathbb{E}_{\nu,\mathrm{HAC}} \left[ \mathbb{I}\{E^{\mathcal{A}} \cap E^\nu\} \sum_{t=1}^{T} \Big( \mathbb{E}_{\nu_{a_\nu^*}}[Y] - \mathbb{E}_{\nu_{A_t^\mathrm{HAC}}}[Y] \Big) \right].
\end{aligned}
\tag{B.4}
$$

Let $\tau_0^\mathrm{HAC}$ be the last round on which the algorithm uniformly explores and $\tau_1^\mathrm{HAC}$ be the last round on which $A_t^\mathrm{HAC} = A_t^\mathrm{C}$. Since the algorithm is deterministic, $\tau_0^\mathrm{HAC}$ is fixed but $\tau_1^\mathrm{HAC}$ is stochastic. Then,

$$
\begin{aligned}
&\mathbb{E}_{\nu,\mathrm{HAC}} \left[ \mathbb{I}\{E^{\mathcal{A}} \cap E^\nu\} \sum_{t=1}^{T} \Big( \mathbb{E}_{\nu_{a_\nu^*}}[Y] - \mathbb{E}_{\nu_{A_t^\mathrm{HAC}}}[Y] \Big) \right] \\
&= \tau_0^\mathrm{HAC} + \mathbb{E}_{\nu,\mathrm{HAC}} \left[ \mathbb{I}\{E^{\mathcal{A}} \cap E^\nu\} \sum_{\tau_0^\mathrm{HAC} < t \leq \tau_1^\mathrm{HAC}} \Big( \mathbb{E}_{\nu_{a_\nu^*}}[Y] - \mathbb{E}_{\nu_{A_t^\mathrm{C}}}[Y] \Big) \right] \\
&\quad + \mathbb{E}_{\nu,\mathrm{HAC}} \left[ \mathbb{I}\{E^{\mathcal{A}} \cap E^\nu\} \sum_{t > \tau_1^\mathrm{HAC}} \Big( \mathbb{E}_{\nu_{a_\nu^*}}[Y] - \mathbb{E}_{\nu_{A_t^\mathrm{UCB}}}[Y] \Big) \right].
\end{aligned}
\tag{B.5}
$$

Suppose that $\nu$ is conditionally benign and $\tilde{\nu}(Z)$ and $\nu(Z)$ are $\varepsilon$-close for $\varepsilon \leq T^{-1/4}\sqrt{|\mathcal{A}|\,|\mathcal{Z}|\log T}$. By triangle inequality, on the event $E^\nu$

$$
\sup_{a \in \mathcal{A}} \sum_{z \in \mathcal{Z}} \left| \mathbb{P}_{\tilde{\nu}_a}[Z = z] - \mathbb{P}_{\hat{\nu}_a}[Z = z] \right| \leq \frac{2\sqrt{|\mathcal{A}|\,|\mathcal{Z}|\log T}}{T^{1/4}},
$$

and thus $\tilde{\nu}(Z)$ will *not* be replaced by $\hat{\nu}(Z)$. Further, on $E^{\mathcal{A}} \cap E^{\mathcal{Z}}$ with $\delta = 2/T^2$, for all $t$ it holds that

$$\mathrm{UCB}^{\mathcal{A}}_{t-1}(a) - \widetilde{\mathrm{UCB}}_{t-1}(a)$$

$$\geq \mathbb{E}_{\nu_a}[Y] - \sum_{z \in \mathcal{Z}} \left( \mathbb{E}_\nu[Y \mid Z = z] + 2\sqrt{\frac{\log T}{\mathbb{T}^{\mathcal{Z}}_{t-1}(z)}} \right) \mathbb{P}_{\tilde{\nu}_a}[Z = z]$$

$$= \sum_{z \in \mathcal{Z}} \mathbb{E}_\nu[Y \mid Z = z]\mathbb{P}_{\nu_a}[Z = z] - \sum_{z \in \mathcal{Z}} \left( \mathbb{E}_\nu[Y \mid Z = z] + 2\sqrt{\frac{\log T}{\mathbb{T}^{\mathcal{Z}}_{t-1}(z)}} \right) \mathbb{P}_{\tilde{\nu}_a}[Z = z]$$

$$\geq -2\sum_{z \in \mathcal{Z}} \sqrt{\frac{\log T}{\mathbb{T}^{\mathcal{Z}}_{t-1}(z)}}\mathbb{P}_{\tilde{\nu}_a}[Z = z] - \frac{\sqrt{|\mathcal{A}|\,|\mathcal{Z}|\log T}}{T^{1/4}}$$

and

$$\mathrm{UCB}^{\mathcal{A}}_{t-1}(a) - \widetilde{\mathrm{UCB}}_{t-1}(a)$$

$$\leq \mathbb{E}_{\nu_a}[Y] + 2\sqrt{\frac{\log T}{\mathbb{T}^{\mathcal{A}}_{t-1}(a)}} - \sum_{z \in \mathcal{Z}} \mathbb{E}_\nu[Y \mid Z = z]\mathbb{P}_{\tilde{\nu}_a}[Z = z]$$

$$= \sum_{z \in \mathcal{Z}} \mathbb{E}_\nu[Y \mid Z = z]\mathbb{P}_{\nu_a}[Z = z] + 2\sqrt{\frac{\log T}{\mathbb{T}^{\mathcal{A}}_{t-1}(a)}} - \sum_{z \in \mathcal{Z}} \mathbb{E}_\nu[Y \mid Z = z]\mathbb{P}_{\tilde{\nu}_a}[Z = z]$$

$$\leq 2\sqrt{\frac{\log T}{\mathbb{T}^{\mathcal{A}}_{t-1}(a)}} + \frac{\sqrt{|\mathcal{A}|\,|\mathcal{Z}|\log T}}{T^{1/4}},$$

and hence $\tau^{\mathrm{HAC}}_1 = T$. Thus, by Lemma B.2 with $\delta = 2/T^2$, we can actually bound Eq. (B.5) using

$$\mathbb{E}_{\nu,\mathrm{HAC}}\left[ \mathbb{I}\{E^{\mathcal{A}} \cap E^\nu\} \sum_{t=1}^T \left( \mathbb{E}_{\nu_{a^*_\nu}}[Y] - \mathbb{E}_{\nu_{A^{\mathrm{HAC}}_t}}[Y] \right) \right]$$

$$\leq \tau^{\mathrm{HAC}}_0 + 2|\mathcal{Z}| + \mathbb{E}_{\nu,\mathrm{HAC}}\left[ \mathbb{I}\{E^{\mathcal{A}} \cap E^{\mathcal{Z}} \cap E^\nu\} \sum_{t=\tau^{\mathrm{HAC}}_0+1}^T \left( \mathbb{E}_{\nu_{a^*_\nu}}[Y] - \mathbb{E}_{\nu_{A^{\mathrm{HAC}}_t}}[Y] \right) \right]$$

$$\leq \tau^{\mathrm{HAC}}_0 + 2|\mathcal{Z}| + \mathbb{E}_{\nu,\mathrm{C}}\left[ \mathbb{I}\{E^{\mathcal{A}} \cap E^{\mathcal{Z}}\} \sum_{t=1}^T \left( \mathbb{E}_{\nu_{a^*_\nu}}[Y] - \mathbb{E}_{\nu_{A^{\mathrm{C}}_t}}[Y] \right) \right]$$

$$\leq 5\sqrt{T} + 2|\mathcal{Z}| + 4\sqrt{2|\mathcal{Z}|\,T\log T} + (\log T)\sqrt{2T} + 4\sqrt{\log T} + \varepsilon(2 + 2\sqrt{\log T})T,$$

where the last line follows from Eqs. (B.2) and (B.3) and the definition of $\tau^{\mathrm{HAC}}_0$. The statement follows from combining this with Eq. (B.4).

Otherwise, consider when $\nu$ is not conditionally benign. By Theorem 4.1,

$$\mathbb{E}_{\nu,\mathrm{HAC}}\left[ \mathbb{I}\{E^{\mathcal{A}} \cap E^\nu\} \sum_{t>\tau^{\mathrm{HAC}}_1} \left( \mathbb{E}_{\nu_{a^*_\nu}}[Y] - \mathbb{E}_{\nu_{A^{\mathrm{HAC}}_t}}[Y] \right) \right]$$

$$\leq \mathbb{E}_{\nu,\mathrm{UCB}}\left[ \mathbb{I}\{E^{\mathcal{A}}\} \sum_{t=1}^T \left( \mathbb{E}_{\nu_{a^*_\nu}}[Y] - \mathbb{E}_{\nu_{A^{\mathrm{UCB}}_t}}[Y] \right) \right] \tag{B.6}$$

$$\leq 4\sqrt{2|\mathcal{A}|\,T\log T}.$$

It remains to focus on the regret contribution from when $A^{\mathrm{HAC}}_t = A^{\mathrm{C}}_t$. The key observation is that if $t \leq \tau^{\mathrm{HAC}}_1$ then the hypothesis test passed for this round. We decompose the regret incurred using

$$\mathbb{E}_{\nu,\mathrm{HAC}}\left[ \mathbb{I}\{E^{\mathcal{A}} \cap E^\nu\} \sum_{\tau^{\mathrm{HAC}}_0 < t \leq \tau^{\mathrm{HAC}}_1} \left( \mathbb{E}_{\nu_{a^*_\nu}}[Y] - \mathbb{E}_{\nu_{A^{\mathrm{C}}_t}}[Y] \right) \right]$$

$$= \mathbb{E}_{\nu,\mathrm{HAC}}\left[ \mathbb{I}\{E^{\mathcal{A}} \cap E^\nu\} \sum_{\tau^{\mathrm{HAC}}_0 < t \leq \tau^{\mathrm{HAC}}_1} \left( \mathbb{E}_{\nu_{a^*_\nu}}[Y] - \widetilde{\mathrm{UCB}}_{t-1}(A^{\mathrm{C}}_t) + \widetilde{\mathrm{UCB}}_{t-1}(A^{\mathrm{C}}_t) - \mathbb{E}_{\nu_{A^{\mathrm{C}}_t}}[Y] \right) \right].$$

First, on the event $E^{\mathcal{A}} \cap E^{\nu}$ with $\delta = 2/T^2$,

$$\sum_{\tau_0^{\mathrm{HAC}} < t \leq \tau_1^{\mathrm{HAC}}} \left( \mathbb{E}_{\nu_{a_\nu^*}}[Y] - \widetilde{\mathrm{UCB}}_{t-1}(A_t^{\mathrm{C}}) \right)$$

$$\leq_{(a)} \sum_{\tau_0^{\mathrm{HAC}} < t \leq \tau_1^{\mathrm{HAC}}} \left( \mathrm{UCB}_{t-1}^{\mathcal{A}}(a_\nu^*) - \widetilde{\mathrm{UCB}}_{t-1}(A_t^{\mathrm{C}}) \right)$$

$$\leq_{(b)} \sum_{\tau_0^{\mathrm{HAC}} < t \leq \tau_1^{\mathrm{HAC}}} \left( \widetilde{\mathrm{UCB}}_{t-1}(a_\nu^*) - \widetilde{\mathrm{UCB}}_{t-1}(A_t^{\mathrm{C}}) + 2\sqrt{\frac{\log T}{\mathbb{T}_{t-1}^{\mathcal{A}}(a_\nu^*)}} \right) + 2T^{3/4}\sqrt{|\mathcal{A}|\,|\mathcal{Z}| \log T}$$

$$\leq_{(c)} 2 \sum_{\tau_0^{\mathrm{HAC}} < t \leq \tau_1^{\mathrm{HAC}}} \sqrt{\frac{\log T}{\mathbb{T}_{t-1}^{\mathcal{A}}(a_\nu^*)}} + 2T^{3/4}\sqrt{|\mathcal{A}|\,|\mathcal{Z}| \log T}$$

$$\leq_{(d)} 2 \sum_{\tau_0^{\mathrm{HAC}} < t \leq \tau_1^{\mathrm{HAC}}} \sqrt{\frac{|\mathcal{A}| \log T}{\sqrt{T}}} + 2T^{3/4}\sqrt{|\mathcal{A}|\,|\mathcal{Z}| \log T}$$

$$\leq 4\,T^{3/4}\sqrt{|\mathcal{A}|\,|\mathcal{Z}| \log T},$$

(B.7)

where we have used that (a) $E^{\mathcal{A}}$ holds; (b) the hypothesis test for HAC-UCB passed; (c) $A_t^{\mathrm{C}} = \arg\max_{a \in \mathcal{A}} \widetilde{\mathrm{UCB}}_{t-1}(a)$; and (d) for $\tau_0^{\mathrm{HAC}} < t$, $\mathbb{T}_{t-1}^{\mathcal{A}}(a_\nu^*) \geq \mathbb{T}_{\tau_0^{\mathrm{HAC}}}^{\mathcal{A}}(a_\nu^*) \geq \sqrt{T}/|\mathcal{A}|$.

Second, again on the event $E^{\mathcal{A}} \cap E^{\nu}$ with $\delta = 2/T^2$,

$$\sum_{\tau_0^{\mathrm{HAC}} < t \leq \tau_1^{\mathrm{HAC}}} \left( \widetilde{\mathrm{UCB}}_{t-1}(A_t^{\mathrm{C}}) - \mathbb{E}_{\nu_{A_t^{\mathrm{C}}}}[Y] \right)$$

$$\leq_{(a)} \sum_{\tau_0^{\mathrm{HAC}} < t \leq \tau_1^{\mathrm{HAC}}} \left( \mathrm{UCB}_{t-1}^{\mathcal{A}}(A_t^{\mathrm{C}}) + 2\sum_{z \in \mathcal{Z}} \sqrt{\frac{\log T}{\mathbb{T}_{t-1}^{\mathcal{Z}}(z)}} \mathbb{P}_{\tilde{\nu}_a}[Z = z] + \frac{2\sqrt{|\mathcal{A}|\,|\mathcal{Z}| \log T}}{T^{1/4}} - \mathbb{E}_{\nu_{A_t^{\mathrm{C}}}}[Y] \right)$$

$$\leq_{(b)} \sum_{\tau_0^{\mathrm{HAC}} < t \leq \tau_1^{\mathrm{HAC}}} \left( 2\sqrt{\frac{\log T}{\mathbb{T}_{t-1}^{\mathcal{A}}(A_t^{\mathrm{HAC}})}} + 2\sum_{z \in \mathcal{Z}} \sqrt{\frac{\log T}{\mathbb{T}_{t-1}^{\mathcal{Z}}(z)}} \mathbb{P}_{\tilde{\nu}_a}[Z = z] + \frac{\sqrt{|\mathcal{A}|\,|\mathcal{Z}| \log T}}{T^{1/4}} \right)$$

$$\leq_{(c)} \sum_{\tau_0^{\mathrm{HAC}} < t \leq \tau_1^{\mathrm{HAC}}} \left( 2\sqrt{\frac{\log T}{\mathbb{T}_{t-1}^{\mathcal{A}}(A_t^{\mathrm{HAC}})}} + 2\sum_{z \in \mathcal{Z}} \sqrt{\frac{\log T}{\mathbb{T}_{t-1}^{\mathcal{Z}}(z)}} \mathbb{P}_{\nu_a}[Z = z] + 7\frac{(\log T)\sqrt{|\mathcal{A}|\,|\mathcal{Z}|}}{T^{1/4}} \right)$$

$$\leq_{(d)} 4\sqrt{2\,|\mathcal{A}|\,T \log T} + 7T^{3/4}(\log T)\sqrt{|\mathcal{A}|\,|\mathcal{Z}|} + 2 \sum_{\tau_0^{\mathrm{HAC}} < t \leq \tau_1^{\mathrm{HAC}}} \sum_{z \in \mathcal{Z}} \sqrt{\frac{\log T}{\mathbb{T}_{t-1}^{\mathcal{Z}}(z)}} \mathbb{P}_{\nu_a}[Z = z],$$

(B.8)

where we have used that (a) the hypothesis test for HAC-UCB passed; (b) $E^{\mathcal{A}}$ holds; (c) $\tilde{\nu}(Z)$ and $\nu(Z)$ are $3T^{-1/4}\sqrt{|\mathcal{A}|\,|\mathcal{Z}| \log T}$-close (if the original $\tilde{\nu}$ does not satisfy this, then by triangle inequality it is replaced with $\hat{\nu}$ that does satisfy this); and (d) Lemma B.5. It remains to take expectation and apply Lemma B.6, and then combine Eqs. (B.6) to (B.8) to obtain

$$
\begin{aligned}
R_{\nu,\mathrm{HAC}}(T) \leq\ & 4\,|\mathcal{A}| + 4\sqrt{2\,|\mathcal{A}|\,T \log T} + 5\sqrt{T} + 4\,T^{3/4}\sqrt{|\mathcal{A}|\,|\mathcal{Z}| \log T} \\
& + 4\sqrt{2\,|\mathcal{A}|\,T \log T} + 7T^{3/4}(\log T)\sqrt{|\mathcal{A}|\,|\mathcal{Z}|} \\
& + 2\sqrt{\log T}\left[ \sqrt{8\,|\mathcal{Z}|\,T} + \sqrt{(T/2)\log T} + 2 \right].
\end{aligned}
$$

(B.9)

$\square$

## B.6 Proof of Theorem 4.8

Supposing the event $E^{\mathcal{A}}$ holds (from Lemma B.1), every $a \in \mathcal{A}_0$ satisfies

$$\text{UCB}_{t-1}^{\mathcal{A}}(a) \geq \mu^{\mathcal{A},\mathcal{Z}}(0,1)p^{\mathcal{A}}(0) + \mu^{\mathcal{A},\mathcal{Z}}(0,0)(1 - p^{\mathcal{A}}(0)).$$

We use the same environment construction from the proof of Theorem 4.5 in Appendix B.4. Using the specific choice of $\delta_T$, this argument implies that on the event $F \cap G$, $A_t^{\mathsf{C}} \in \mathcal{A}_1$ for sufficiently large $T$ and $t \geq 2\sqrt{T}$. Recall that

$$\hat{\mu}_t^{\mathcal{Z}}(z) = \frac{\mathbb{T}_t^{\mathcal{A},\mathcal{Z}}(0,z)}{\mathbb{T}_t^{\mathcal{Z}}(z)}\hat{\mu}_t^{\mathcal{A},\mathcal{Z}}(0,z) + \frac{\mathbb{T}_t^{\mathcal{A},\mathcal{Z}}(1,z)}{\mathbb{T}_t^{\mathcal{Z}}(z)}\hat{\mu}_t^{\mathcal{A},\mathcal{Z}}(1,z).$$

Since $\mathbb{T}_t^{\mathcal{A},\mathcal{Z}}(0,z) \leq 2\sqrt{T}$ and $\mathbb{T}_t^{\mathcal{Z}}(z)$ grows linearly in $t$ on the event $F \cap G$, for every $\alpha \in (0,1)$ and $\varepsilon > 0$ there exists $T$ large enough such that for all $t \geq (\log T)\sqrt{T}$ (crucially, the $\log T$ ensures that the proportion of these $t$ where $A_t^{\mathsf{C}} \in \mathcal{A}_1$ tends to 1 as $T$ gets larger) it holds that for $z \in \mathcal{Z}_j$,

$$\hat{\mu}_t^{\mathcal{Z}}(z) \leq \alpha\mu^{\mathcal{A},\mathcal{Z}}(0,j) + (1-\alpha)\mu^{\mathcal{A},\mathcal{Z}}(1,j) + \varepsilon.$$

This implies that for all $\varepsilon > 0$, taking $\alpha$ small enough and $T$ large enough with $t \geq (\log T)\sqrt{T}$ gives

$$\widetilde{\text{UCB}}_{t-1}(a) \leq \mu^{\mathcal{A},\mathcal{Z}}(1,1)p^{\mathcal{A}}(0) + \mu^{\mathcal{A},\mathcal{Z}}(1,0)(1 - p^{\mathcal{A}}(0)) + \varepsilon.$$

By the exploration phase,

$$2\sqrt{\frac{\log T}{\mathbb{T}_{t-1}^{\mathcal{A}}(a)}} \leq 2\sqrt{\frac{|\mathcal{A}|\log T}{\sqrt{T}}},$$

and hence can be made arbitrarily small by taking $T$ sufficiently large. Thus, under the assumption

$$[\mu^{\mathcal{A},\mathcal{Z}}(0,1) - \mu^{\mathcal{A},\mathcal{Z}}(1,1)]p^{\mathcal{A}}(0) + [\mu^{\mathcal{A},\mathcal{Z}}(0,0) - \mu^{\mathcal{A},\mathcal{Z}}(1,0)](1 - p^{\mathcal{A}}(0)) > 0, \tag{B.10}$$

for large enough $T$ the first condition from Algorithm 1 will fail for some $a \in \mathcal{A}_0$ when $t = (\log T)\sqrt{T}$. Note that Eq. (B.10) is satisfied by the example given in the proof of Theorem 4.5. This implies that, on the event $F \cap G \cap E^{\mathcal{A}}$, for large enough $T$ the HAC-UCB algorithm switches to following UCB when $t = (\log T)\sqrt{T}$. Since this joint event holds with probability larger than $1 - 2(|\mathcal{A}| + |\mathcal{Z}|)/T$, combining the exploration phase regret with the regret bound of Theorem 4.1 gives the result. $\qquad\square$

## C Proof of Theorem 6.2

Fix $\mathcal{A}$, $\mathcal{Z}$, and $T$. Let $\mathcal{Z}_0$ be an arbitrary strict subset of $\mathcal{Z}$ and $\mathcal{Z}_1 = \mathcal{Z} \setminus \mathcal{Z}_0$. Fix $\Delta \in (0, 1/20)$ and $\varepsilon \in (0,1)$ to be chosen later. Define the family of marginal distributions

$$q_a[Z \in \mathcal{Z}_0] = \begin{cases} 1/2 + 2\Delta & a = 1 \\ 1/2 & a \neq 1, \end{cases}$$

where probability is evenly spaced within $\mathcal{Z}_0$ and $\mathcal{Z}_1$ respectively. Further, define the Bernoulli conditional response distribution

$$\tilde{p}[Y = 1 \mid Z = z] = \begin{cases} 3/4 & z \in \mathcal{Z}_0 \\ 1/4 & z \in \mathcal{Z}_1. \end{cases}$$

Define the Bernoulli conditionally benign environment $\tilde{\nu} \in \mathscr{P}(\mathcal{Z} \times \mathcal{Y})^{\mathcal{A}}$ for all $a \in \mathcal{A}$ by

$$\mathbb{P}_{\tilde{\nu}_a}[Y = 1] = \sum_{z \in \mathcal{Z}} \tilde{p}[Y = 1 \mid Z = z]q_a[Z = z].$$

Notice that

$$\mathbb{E}_{\tilde{\nu}_a}[Y] = \begin{cases} 1/2 + \Delta & a = 1 \\ 1/2 & a \neq 1. \end{cases}$$

Then, for every $a_0 \neq 1$, define $\nu^{a_0} \in \mathscr{P}(\mathcal{Z} \times \mathcal{Y})^{\mathcal{A}}$ for all $a \in \mathcal{A}$ by

$$\mathbb{P}_{\nu_a^{a_0}}[Y = 1] = \sum_{z \in \mathcal{Z}} p_a^{a_0}[Y = 1 \mid Z = z] q_a[Z = z],$$

where

$$p_a^{a_0}[Y = 1 \mid Z = z] = \begin{cases} 3/4 & a = 1, z \in \mathcal{Z}_0 \\ 1/4 & a = 1, z \in \mathcal{Z}_1 \\ 3/4 + 2\Delta(1 + \varepsilon) & a = a_0, z \in \mathcal{Z}_0 \\ 1/4 & a = a_0, z \in \mathcal{Z}_1 \\ 3/4 & a \notin \{1, a_0\}, z \in \mathcal{Z}_0 \\ 1/4 & a \notin \{1, a_0\}, z \in \mathcal{Z}_1. \end{cases}$$

Notice that $\nu^{a_0}$ is *not* conditionally benign, and

$$\mathbb{E}_{\nu_a^{a_0}}[Y] = \begin{cases} 1/2 + \Delta & a = 1 \\ 1/2 + \Delta(1 + \varepsilon) & a = a_0 \\ 1/2 & a \notin \{1, a_0\}. \end{cases}$$

We now extend Lemma 15.1 of Lattimore and Szepesvári [19]. In particular, let $\pi^q = \mathfrak{a}(\mathcal{A}, \mathcal{Z}, q, T)$ and observe that for any $\nu \in \Pi_{\mathcal{A}, \mathcal{Z}}(q)$,

$$d\mathbb{P}_{\nu,\pi^q}(H_T) = \prod_{t=1}^{T} \pi_t^q(A_t \mid H_{t-1}) \mathbb{P}_\nu[Z_t(A_t), Y_t(A_t) \mid A_t].$$

Thus, for every $a_0 \neq 1$,

$$\begin{aligned}
\mathrm{KL}\left(\mathbb{P}_{\tilde{\nu},\pi^q} \,\|\, \mathbb{P}_{\nu^{a_0},\pi^q}\right) &= \mathbb{E}_{\tilde{\nu},\pi^q}\left[\log \frac{d\mathbb{P}_{\tilde{\nu},\pi^q}}{d\mathbb{P}_{\nu^{a_0},\pi^q}}(H_T)\right] \\
&= \sum_{t=1}^{T} \mathbb{E}_{\tilde{\nu},\pi^q}\left[\mathbb{E}_{\tilde{\nu},\pi^q}\left[\log \frac{\mathbb{P}_{\tilde{\nu}}[Z_t(A_t), Y_t(A_t)]}{\mathbb{P}_{\nu^{a_0}}[Z_t(A_t), Y_t(A_t)]} \,\Big|\, A_t\right]\right] \\
&= \sum_{t=1}^{T} \mathbb{E}_{\tilde{\nu},\pi^q}\left[\mathrm{KL}\left(\mathbb{P}_{\tilde{\nu}_{A_t}} \,\|\, \mathbb{P}_{\nu_{A_t}^{a_0}}\right)\right] \\
&= \sum_{a \in \mathcal{A}} \mathbb{E}_{\tilde{\nu},\pi^q}[\mathbb{T}_T^{\mathcal{A}}(a)] \, \mathrm{KL}\left(\mathbb{P}_{\tilde{\nu}_a} \,\|\, \mathbb{P}_{\nu_a^{a_0}}\right).
\end{aligned}$$

Since the marginal distribution $q$ is shared, for each $a \in \mathcal{A}$ this simplifies to

$$\begin{aligned}
\mathrm{KL}\left(\mathbb{P}_{\tilde{\nu}_a} \,\|\, \mathbb{P}_{\nu_a^{a_0}}\right) &= \sum_{z \in \mathcal{Z}} q_a(Z = z) \mathrm{KL}\left(\tilde{\nu}_a(Y \mid z) \,\|\, \nu_a^{a_0}(Y \mid z)\right) \\
&= \begin{cases} 0 & a \neq a_0 \\ (1/2)\mathrm{KL}\left(\mathrm{Ber}(3/4) \,\|\, \mathrm{Ber}(3/4 + 2\Delta(1 + \varepsilon))\right) & a = a_0. \end{cases}
\end{aligned}$$

Thus, by Pinsker's inequality (Theorem 14.2 of [19]),

$$\begin{aligned}
R_{\tilde{\nu},\pi^q}(T) + R_{\nu^{a_0},\pi^q}(T) &> \frac{T\Delta}{2}\mathbb{P}_{\tilde{\nu},\pi^q}[\mathbb{T}_T^{\mathcal{A}}(1) \leq T/2] + \frac{T\Delta\varepsilon}{2}\mathbb{P}_{\nu^{a_0},\pi^q}[\mathbb{T}_T^{\mathcal{A}}(1) > T/2] \\
&\geq \frac{T\Delta\varepsilon}{4}\exp\{-\mathrm{KL}\left(\mathbb{P}_{\tilde{\nu},\pi^q} \,\|\, \mathbb{P}_{\nu^{a_0},\pi^q}\right)\} \\
&= \frac{T\Delta\varepsilon}{4}\exp\left\{-\mathbb{E}_{\tilde{\nu},\pi^q}[\mathbb{T}_T^{\mathcal{A}}(a_0)]\,(1/2)\mathrm{KL}\left(\mathrm{Ber}(3/4) \,\|\, \mathrm{Ber}(3/4 + 2\Delta(1 + \varepsilon))\right)\right\}.
\end{aligned}$$

Using that $\mathrm{KL}\left(\mathrm{Ber}(3/4) \,\|\, \mathrm{Ber}(3/4 + 2\Delta(1 + \varepsilon))\right) \leq 4x^2$ for $x < 1/10$ and the assumption of the theorem, this implies that for all $T$,

$$\mathbb{E}_{\tilde{\nu},\pi}[\mathbb{T}_T^{\mathcal{A}}(a_0)] \geq \frac{\log(T\Delta\varepsilon) - \log(8C\sqrt{|\mathcal{A}|T})}{(1/2)\mathrm{KL}\left(\mathrm{Ber}(3/4) \,\|\, \mathrm{Ber}(3/4 + 2\Delta(1 + \varepsilon))\right)} \geq \frac{1}{8\Delta^2(1 + \varepsilon)^2}\log \frac{\Delta\varepsilon\sqrt{T}}{8C\sqrt{|\mathcal{A}|}}.$$

Finally, we combine this with

$$R_{\tilde{\nu}, \pi}(T) = \sum_{a_0 \neq 1} \Delta \mathbb{E}_{\tilde{\nu}, \pi}[\mathbb{T}_T^{\mathcal{A}}(a_0)],$$

choose $\varepsilon = 1$, and set

$$\Delta = \frac{16C \sqrt{|\mathcal{A}|}}{\sqrt{T}}.$$

$\square$

## D Simulation Details

Here we provide more details for the simulations in Section 5. First, the regret bounds are computed by sampling a new data realization for each horizon $T$ we consider, computing the expected regret (with respect to the data randomness) for this realization, and then averaging this value (i.e., over the algorithm randomness) over $M = 300$ realizations.

C-UCB and UCB are implemented exactly according to Subsection 2.2, HAC-UCB is implemented exactly according to Algorithm 1, and C-UCB-2 is implemented exactly according to Algorithm 3 of Nair et al. [27] (including their time-adaptive confidence bound). For Corral, we use the log-barrier method from Algorithms 1 and 2 of Agarwal et al. [2] with base algorithms UCB and C-UCB. We use the prescribed learning rate from their Theorem 5 of

$$\eta = \frac{1}{40 \cdot \mathcal{R}(T) \log T},$$

where $\mathcal{R}(T)$ is an upper bound on the regret of C-UCB. In order to use UCB and C-UCB with importance-weighted losses, we implement the epoch-based approach of Arora et al. [4] along with their Freedman's inequality confidence bound of

$$\sqrt{\frac{4\rho \log t}{\mathbb{T}_t}} + \frac{4\rho \log t}{3\mathbb{T}_t}$$

for the arm means (UCB) and the post-action context conditional means (C-UCB) respectively, where $\rho$ is an upper bound on the importance-weighted losses.

For Corral, it is typical in experiments (e.g., [4]) to swap out the Freedman's inequality confidence bound for the usual Hoeffding's inequality confidence bound. However, there are no theoretical guarantees for the algorithm then (due to the importance-weighted losses), and we observed in additional experiments that it is still not adaptive (i.e., it does no better than UCB even in conditionally benign environments).