# OpenReview forum: "Adaptively Exploiting d-Separators with Causal Bandits"
_NeurIPS.cc/2022/Conference — NeurIPS 2022 Accept_

### Official Review · Reviewer_GygT · 2022-07-10

**Rating:** 7
**Confidence:** 3
**Soundness:** 3 good
**Presentation:** 3 good
**Contribution:** 3 good

**Summary:**

This paper studies the problem of online learning in causal bandits, i.e., multi-armed bandit models compatible with a certain set of structural constraints encoded in a causal graph. Existing methods assume that the correct causal graph is provided. This paper studies an important generalization of this problem setting where the causal graph could be misspecified. The authors propose an algorithm that could achieve a sub-linear regret in the most general cases where the graph is incorrect. It is also able to exploit the existence of independence constraints to improve the learning performance when the graph is accurate. The authors further provide the worst-case analysis showing that any online learning algorithm is unable to achieve the optimal regret bound with and without the causal graph simultaneously when the graph misspecification is possible.

**Questions:**

In Definition 6.1, the constant C in Eqs 6.1 and 6.2 appear to be the same. While in Theorem 6.2, the negative result utilizes two different constraints C and C'. Is this a typo? Otherwise, Theorem 6.2 does not seem to show that "it is impossible to be adaptively minimax optimal with respect to the conditionally benign property."

**Limitations:**

The authors have clearly stated the assumptions behind the proposed methods. This work is mainly theoretical, and its long-term societal impact is unclear to see.

**Strengths And Weaknesses:**

This paper is clearly written and well-organized. It attempts to address a challenging problem: online learning combined with hypothesis testing on pre-specified domain knowledge, which could possibly be misspecified. Existing methods in the causal bandit literature mostly assume the correct causal graph is provided. Therefore, I think this paper could have an impact on both reinforcement learning and causal inference. The proposed algorithm, HAC-UCB, appears sound and demonstrates some desirable properties. That is, the algorithm is able to achieve a sub-linear regret in the most general bandit model while improving the regret bound by exploiting underlying independence constraints when assumptions in the graph are accurate, represented in the form of a set of d-separators.

While it is sub-linear, the regret bound of HAC-UCB is still worst than UCB in the general case. To understand this observation, the authors also provide the worst-case analysis of this learning setting. The result is interesting. It suggests that there exists no online learning algorithm that could achieve the optimal regret bound with and without knowing the correct causal graph simultaneously.  This suggests that the performance gap between HAC-UCB and UCB might be expected.

As for the weakness, I think the lower bound result in Theorem 6.2 does not suggest that the regret of HAC-UCB could not be improved. Theorem 6.2 states that one could not design an online algorithm achieving regret $\mathcal{O}(\sqrt{|A|T})$ (without graph) and $\mathcal{O}(\sqrt{|Z|T})$ (with graph) simultaneously. However, this does not mean that it is infeasible to design a robust online algorithm capable of graph misspecification while achieving the same regret as UCB, which is $\mathcal{O}(\sqrt{|A|T\log(T)})$. The main reason is that the regret bound of UCB is not tight, containing a log factor $log(T)$. Therefore, the authors comment, "In Theorem 6.2, we will show that, up to logarithmic factors, it is impossible to improve the $T^{3/4}$ n this bound to $\sqrt{T}$ while still achieving the improved regret on conditionally benign environments", is somewhat misleading, and should be removed.

---

> ### Author Response · Authors · 2022-08-02
> **Response to Reviewer GygT**
>
> Thank you for your detailed review. We now address your main concern.
>
> ## Interpretation of Theorem 6.2.
>
> **First, we clarify that there is no typo in the quantification of constants, and yet our conclusion that adaptive minimax optimality is impossible is still correct.** To see this, note that for the conditionally benign setting, the RHS of Eqn (6.2) depends on |Z| rather than |A|, and hence since C’ in Theorem 6.2 must be chosen independently of both of these, no algorithm can be adaptively minimax optimal (even if C’ is arbitrarily smaller than the C in the first equation of Theorem 6.2).
>
> Second, you rightfully point out that the dependence on log factors is not fully resolved. Specifically, we do not know if it is possible to obtain “near optimal adaptivity” with respect to log factors in the causal setting, and agree that this is a very interesting open problem to pursue. We will be sure to update the remarks following Theorem 6.2 to make this absolutely clear.
>
> We make no claims that HAC-UCB cannot be improved upon, only that the optimal notion of adaptivity is impossible and hence the HAC-UCB regret cannot be fully improved to the optimal rates respectively (admittedly, we used confusing wording in the sentence you highlighted). In the worst-case, there need not be any dependence on log factors (see, e.g., the Bandit Algorithms book for how to shave these off), and hence our conclusion that adaptive minimax optimality is impossible still holds. We would like to note that it was not even known conclusively until Neurips 2021 (https://proceedings.neurips.cc/paper/2021/hash/49ef08ad6e7f26d7f200e1b2b9e6e4ac-Abstract.html) that the log factors are necessary for UCB, and hence a lot remains to be done in the study of log factors for bandit algorithms, and we believe it is reasonable to defer establishing tight log factors to future work.

---

> > ### Comment · Reviewer_GygT · 2022-08-08
> > **Thank you for the response**
> >
> > I appreciate the authors' response, and some of my concerns have been addressed. This paper studies a novel problem that concerns the trade-off between exploiting (possibly misspecified) graphical structure in specific cases while maintaining a reasonable performance in general cases. The analysis of adaptive minimax optimality in Theorem 6.2 is particularly interesting. Indeed, the regret bound of HAC-UCB might not be optimal and could be improved. However, no one should expect a novel problem to start and end with a single paper. This paper provides an initial baseline for an important online learning problem with uncertain structural assumptions. Due to these reasons, I maintain my assessment and vote for acceptance.

---

### Official Review · Reviewer_tTqa · 2022-07-10

**Rating:** 7
**Confidence:** 4
**Soundness:** 4 excellent
**Presentation:** 3 good
**Contribution:** 4 excellent

**Summary:**

The authors study adaptivity in the setting of causal bandits. They consider a bandit setting where after an action a is taken, a post-action context Z_a is observed. They propose a new property, a conditionally benign bandit environment that subsumes Z being a d-separator and other assumptions previously studied in the literature. They develop an algorithm that is adaptive to whether the bandit environment is conditionally benign. This allows the algorithm to obtain a regret bound matching C-UCB, a SOTA algorithm for causal bandits, when the bandit environment is conditionally benign and that has a sublinear regret otherwise. The authors also show that it is impossible to be minimax optimal at once for environments satisfying the conditionally benign property and violating the conditionally benign property.

**Questions:**

What are real-world examples of the protocol?

Do the authors believe the algorithm could make improvements on real-world problems? It would be nice to see the empirical performance of the algorithm on real-world problems or larger-scale simulations.

**Limitations:**

Yes.

**Strengths And Weaknesses:**

Strengths:

The paper initiates the study of adaptivity in causal bandits, a very important problem.

The authors give a set of strong and novel results. In particular, their impossibility result on strict adaptivity is interesting.

The proposed notion of conditionally benign environments seems like a nice and simple generalization of prior assumptions in the literature.

The authors give a very strong empirical validation of the theory. It is nice to see that existing algorithms do not adapt to the distinct settings, while the proposed algorithm does.

Weaknesses:

The authors do not seem to give real-world examples of the bandit protocol, which may make it seem more like a mathematically convenient setting to study adaptivity in causal bandits. In what applications, would a post-action context be observed?

While the problem itself it important and nice theoretical progress has been made, it is unclear to me that the proposed algorithm would lead to practical gains on real-world problems. In the particular, the initial burn-in time of playing each arm $O(\sqrt{T}/|A|)$ times seems costly and the hypothesis test seems potentially loose. It would be nice to see the empirical performance of the algorithm on real-world problems or larger-scale simulations.

The algorithm uses fairly standard ideas in the literature. It uses a hypothesis test to determine whether the environment is conditionally benign. Therefore, the algorithmic novelty seems limited.

As the authors indicate, the rate $T^{3/4}$ may be suboptimal. I'm curious as to whether this result is tight for the algorithm and whether there may be an algorithm with an improved rate.

---

> ### Author Response · Authors · 2022-08-02
> **Response to Reviewer tTqa**
>
> Thank you for your detailed review. We address the three comments that you raise.
>
> ## Real world applicability
> (the following two paragraphs are repeated in our response to reviewer KRsJ)
>
> We agree that connecting our theoretical results to real world applications is important. Due to their recent development (the theoretical interest only began in 2016), causal bandit algorithms have not yet been widely deployed in real world applications as far as we know, and we think that this is an exciting line of inquiry to pursue in future work. First, note that once interventions are possible (which is the case for randomized control trials, and necessary to avoid unverifiable assumptions), the causal setting is _exactly_ the same as the bandit setting. We now motivate a potential setting of interest where our algorithm could be used, and are happy to add this motivation in the camera-ready discussion.
>
> Settings where HAC-UCB will be useful are those where the intervention space is very large and additional information is available. One such setting—which we are admittedly *not* domain experts in—is learning the causal effect of genes on disease phenotypes (e.g., [1]). Here, scientists have the ability to actually intervene via “perturbations” [2], yet choosing where and how to perturb often results in a combinatorially large number of interventions. As a high-level example, in [3] the authors note that the total number of variations (“single nucleotide polymorphisms”, or SNPs) is far too large to exhaustively test, but propose clustering genes into “modules”. These modules (observed via “gene expression probes”) can act as post-action contexts, but the authors note that many causal diagrams are possible (see their Figure 1), with the modules being actual d-separators, only mediators, or potentially causally unrelated. Hence, **using HAC-UCB could allow one to learn causal effects without depending on the infeasible number of SNPs, and yet also not rely on potentially incorrect assumptions about the causal graph.** Obviously, to state this more precisely and actually deploy our algorithm would require collaboration with domain experts, but we are optimistic that our approach can have concrete benefits in such settings.
>
> [1] https://www.ahajournals.org/doi/10.1161/circresaha.114.302904
>
> [2] https://www.broadinstitute.org/genetic-perturbation-platform
>
> [3] https://bmcproc.biomedcentral.com/articles/10.1186/s12919-016-0009-x
>
> ## Algorithm novelty
>
> Contrary to the full-information setting, adaptivity is poorly understood for bandit problems in general. Further, we are the first to consider adaptivity with respect to causal structure. **We are also not aware of existing work that uses hypothesis tests to obtain adaptivity in this way, particularly for bandit feedback problems. Hence, we believe that our algorithm is novel in this regard.** Indeed, we believe that our hypothesis testing approach may be fruitful for adapting in other settings, even when existing aggregation approaches fail (as we have shown experimentally that Corral fails to adapt in our setting).
>
> ## Algorithm optimality
>
> As noted, we have not demonstrated optimality of HAC-UCB, and it is natural to ask whether the T^{3/4} can be improved. In light of (a) our positive results demonstrating that HAC-UCB dominates existing causal bandit algorithms and (b) our negative results demonstrating that optimal adaptivity is impossible, we believe it is reasonable to defer demonstrating full optimality to future work. We are actively pursuing such results, but have not yet solved this problem.

---

### Official Review · Reviewer_KRsJ · 2022-07-12

**Rating:** 8
**Confidence:** 4
**Soundness:** 4 excellent
**Presentation:** 4 excellent
**Contribution:** 4 excellent

**Summary:**

This paper proposes a novel algorithm for causal bandits that achieves “best of both worlds”: comparable performance to the benchmark C-UCB algorithm in conditionally benign environment while achieving a much better performance in the worst-case environment. In the causal bandit setting, the player observes some other post-action contexts after they play some action, rather than observing all contexts before they play an action. Under this setting, they introduce the conditional benign property which is closely tied with d-separability. They show that the benchmark C-UCB algorithm suffers linear regret when the environment is not conditionally benign, and it is impossible to achieve worst-case optimal regret and optimal regret in the conditional benign environment simultaneously. Then, they introduce the notion of Pareto optimality based on the Pareto frontier and show that their new algorithm, HAC-UCB, is Pareto optimal.


**Questions:**

None in addition to the above.

**Limitations:**

The author could discuss more on the limitations of their proposed algorithm, e.g. in which situation does it work well, and in which situation does it work poorly, in theory and in simulation?


**Strengths And Weaknesses:**

In terms of originality, the paper provides several interesting contributons. Firstly, they introduce the idea of a "post-action" context. Secondly, they provide a novel algorithm, HACUCB to solve the post-action context problem with a specific application of causal bandits when observing a d-seperator of d-separators. Perhaps most interestingly is the new impossibility results for previous algorithms (C-UCB) which achieve optimality in the assumption of a conditional benign environment, and provide interesting lower bounds.

Overall the paper is well written and easy to understand. The impossibility results are very surprising, including those about adaptive minimax adaptivity.  As far as the experiments, they are not extensive, however they demonstrate the advantage of HACUCB over CUCB in benign settings effectively.

My only comment is that although this paper is a direct extension to the existing literature Lu et al, it needs more justification to show the significance. For example, I would love to see some real applications of their algorithm.

Overall, I would love to accept the paper based on the amount of work and technical contributions. I think this paper has interesting ideas and the paper is well written.

---

> ### Author Response · Authors · 2022-08-02
> **Response to Reviewer KRsJ**
>
> Thank you for your detailed review. One small part we want to clarify: while we do introduce the notion of adaptive Pareto optimality for this problem, we do not show HAC-UCB achieves this, and instead leave it as an open problem. The rest of your summary, including that we show (a) adaptive optimality is impossible, (b) C-UCB gets linear regret in the worst case, and (c) HAC-UCB recovers optimal regret in the conditionally benign setting and beats C-UCB in the worst case, is accurate. We now address your comment about real world applications.
>
> ## Real world applicability
> (the following two paragraphs are repeated in our response to reviewer tTqa)
>
> We agree that connecting our theoretical results to real world applications is important. Due to their recent development (the theoretical interest only began in 2016), causal bandit algorithms have not yet been widely deployed in real world applications as far as we know, and we think that this is an exciting line of inquiry to pursue in future work. First, note that once interventions are possible (which is the case for randomized control trials, and necessary to avoid unverifiable assumptions), the causal setting is *exactly* the same as the bandit setting. We now motivate a potential setting of interest where our algorithm could be used, and are happy to add this motivation in the camera-ready discussion.
>
> Settings where HAC-UCB will be useful are those where the intervention space is very large and additional information is available. One such setting—which we are admittedly *not* domain experts in—is learning the causal effect of genes on disease phenotypes (e.g., [1]). Here, scientists have the ability to actually intervene via “perturbations” [2], yet choosing where and how to perturb often results in a combinatorially large number of interventions. As a high-level example, in [3] the authors note that the total number of variations (“single nucleotide polymorphisms”, or SNPs) is far too large to exhaustively test, but propose clustering genes into “modules”. These modules (observed via “gene expression probes”) can act as post-action contexts, but the authors note that many causal diagrams are possible (see their Figure 1), with the modules being actual d-separators, only mediators, or potentially causally unrelated. Hence, **using HAC-UCB could allow one to learn causal effects without depending on the infeasible number of SNPs, and yet also not rely on potentially incorrect assumptions about the causal graph.** Obviously, to state this more precisely and actually deploy our algorithm would require collaboration with domain experts, but we are optimistic that our approach can have concrete benefits in such settings.
>
> [1] https://www.ahajournals.org/doi/10.1161/circresaha.114.302904
>
> [2] https://www.broadinstitute.org/genetic-perturbation-platform
>
> [3] https://bmcproc.biomedcentral.com/articles/10.1186/s12919-016-0009-x
>
> ## Limitations
>
> We briefly remark on two settings where there is no benefit to using HAC-UCB: (a) when |A| is already very small or otherwise |Z| is approximately the same size as |A|, and (b) when it seems very unlikely that Z is a d-separator. In both cases, there are basically no benefits to using any causal bandit algorithm, and hence one should just use a worst-case optimal algorithm like UCB. Note that the guarantees for HAC-UCB still apply in these settings, and still prescribe much better performance than C-UCB (especially in case (b)), just that these settings are basically what UCB was designed to be optimal for.

---

> > ### Comment · Reviewer_KRsJ · 2022-08-09
> > **Response**
> >
> > Thanks for the insightful example. I think including something like it in the paper will contribute greatly.

---

### Official Review · Reviewer_XZzq · 2022-07-12

**Rating:** 7
**Confidence:** 3
**Soundness:** 3 good
**Presentation:** 3 good
**Contribution:** 3 good

**Summary:**

The paper addresses the multi-armed bandit problem where post-action contexts are observed and may or may not be a d-separator. When there is no d-separator, many algorithms including the UCB algorithm achieve the optimal regret. On the other hand when there is information on which variable is a d-separator, UCB style algorithms (such as C-UCB) have been proposed that achieve a regret bound that depends on the cardinality of the d-separator variable instead of the number of actions.
However, there is no former algorithm that addresses the case where the learner does not know whether a post-action variable is a d-separator or not. A safe way would be to apply the original UCB algorithm, but this will achieve suboptimal regret in case there actually is a d-separator. On the other hand, authors prove that using the C-UCB algorithm can result in linear regret in case there is no d-separator.
An adaptive algorithm is required to identify the d-separator variables and decide whether to implement original UCB or C-UCB for action selection. Authors propose a new algorithm based on consecutive hypothesis tests that identify whether a variable is a d-separator. The algorithm achieves optimal regret when there actually is a d-separator, and sublinear (O(T^{3/4})) regret when there is no d-separator. Authors also prove that their bound cannot be shaved off by more than T^{1/4} order.


**Questions:**

Were the experiments performed repetively, or was it just one run?


**Limitations:**

I did not find any discussion on the societal impact.

**Strengths And Weaknesses:**

Strength: The authors define the conditional benign property under which the proposed algorithm achieves optimal regret. This property is a weaker assumption than previous works. Also, the proposed algorithm does not require to know the exact marginal distribution of the d-separator variables. The optimal regret is achieved with approximate distributions as well.

---

> ### Author Response · Authors · 2022-08-02
> **Response to Reviewer XZzq**
>
> Thank you for your detailed review. In regards to your question, the experiments were repeated 300 times for each experiment, and the average value is plotted. Since we know the distribution, we can compare to the exact mean reward in the definition of regret, so the only variability to average over is the algorithms’ randomness.

---

### Meta-Review · Area_Chair_KZyp · 2022-08-20

**Recommendation:** Accept
**Confidence:** Certain

**Metareview:**

This paper exploits the causal structure in the multi-armed bandits setting and gives a set of novel and strong results, including (1) the conditional benign property -- a nice and simple generalization of prior assumptions; (2) an impossibility result for the previous algorithm C-UCB; and (3) a new algorithm gives sublinear regret in any cases and optimal regret when there actually is a d-separator.  The paper is well-organized and nicely written.  The reviewers are unanimously positive about this paper.

**Award:**

No

---

### Decision · Program_Chairs · 2022-09-14

Accept